# A new experimental platform facilitates assessment of the transcriptional and chromatin landscapes of aging yeast

**David G Hendrickson, Ilya Soifer, Bernd J Wranik, Griffin Kim, Michael Robles, Patrick A Gibney[†], R Scott McIsaac***

Calico Life Sciences LLC, South San Francisco, United States

**Abstract** Replicative aging of *Saccharomyces cerevisiae* is an established model system for eukaryotic cellular aging. A limitation in yeast lifespan studies has been the difficulty of separating old cells from young cells in large quantities. We engineered a new platform, the Miniature-chemostat Aging Device (MAD), that enables purification of aged cells at sufficient quantities for genomic and biochemical characterization of aging yeast populations. Using MAD, we measured DNA accessibility and gene expression changes in aging cells. Our data highlight an intimate connection between aging, growth rate, and stress. Stress-independent genes that change with age are highly enriched for targets of the signal recognition particle (SRP). Combining MAD with an improved ATAC-seq method, we find that increasing proteasome activity reduces rDNA instability usually observed in aging cells and, contrary to published findings, provide evidence that global nucleosome occupancy does not change significantly with age.
DOI: https://doi.org/10.7554/eLife.39911.001

**\*For correspondence:**
Corresponding Author: rsm@
calicolabs.com

**Present address:** [†]Cornell
University, Ithaca, United States

**Competing interest:** See
page 24

**Reviewing editor:** Matt
Kaeberlein, University of
Washington, United States

## Introduction

Aging is a multifactorial process characterized by loss of homeostasis, reduced fitness, and increased risk of death. In people, aging is the primary risk factor for a myriad of ailments including dementia, cancer, and heart disease (*Niccoli and Partridge, 2012*). Across different organisms, aging is influenced by both environmental (*Chetty et al., 2016*; *Harrison et al., 2009*; *McIsaac et al., 2016*) and genetic factors (*Kenyon, 2010*); defining a quantitative systems-level understanding of the molecular changes that cause aging remains an open problem.

The unicellular eukaryote *Saccharomyces cerevisiae* (budding yeast), which typically produces ~25 daughter cells before dying (*Mortimer and Johnston, 1959*), has enabled several insights into the cellular aging process via a combination of cell biological, genetic, and genomic approaches (*Hughes and Gottschling, 2012*; *McCormick et al., 2015*; *Janssens et al., 2015*; *Hu et al., 2014*; *Defossez et al., 1999*). For example, instability at the ribosomal DNA (rDNA) locus, which contains ~100–200 tandemly repeated copies in the genome, can result in the formation of rDNA extra-chromosomal circles (ERCs), and is is a major determinant of yeast replicative lifespan (RLS) (*Defossez et al., 1999*; *Sinclair and Guarente, 1997*; *Kobayashi et al., 1998*; *Stumpferl et al., 2012*; *Li et al., 2017*). A single-nucleotide polymorphism that reduces the rate of origin firing at the rDNA locus increases rDNA stability, and thereby increases RLS (*Kwan et al., 2013*; *Foss et al., 2017*). In a seemingly unrelated pathway, a decrease in vacuolar acidity early in a mother cell's life promotes mitochondrial dysfunction and limits her lifespan (*Hughes and Gottschling, 2012*). Likewise, caloric restriction-mediated life extension, acting through the homeostatic regulator *GCN4*, is a well-studied yeast RLS intervention that is broadly phenocopied by perturbations that inhibit translation (*Steffen et al., 2008*; *Mittal et al., 2017*). In a step towards a systems-level understanding of the genetic determinants of cellular aging, RLSs of >4500 gene deletions were recently measured

using microdissection to physically separate daughter cells from mother cells and manually counting the number of divisions before mother cell death (*McCormick et al., 2015*). Even still, no unifying model has coalesced from these disparate observations. The rarity of old mother cells in mitotically growing populations and the large number of cells required for genomic and biochemical assays has hampered our ability to dynamically monitor genomic changes in aging cells that would aid in building and testing explanatory models of aging.

With currently available technology, obtaining large numbers of old mother cells requires physical separation by magnetic cell sorting, which can be combined with daughter-cell-killing genetic programs to increase the average age and purity of purified populations (*Lindstrom and Gottschling, 2009*; *Patterson et al., 2015*). Profiling aging populations *densely* across multiple ages or broadly across genotype and environment can be improved with fluidic approaches (*Janssens et al., 2015*). To our knowledge, no genomic assay (RNA-seq, ATAC-seq, etc.) has been published on strains harboring longevity-associated mutations as they age. Identifying molecular changes, and perhaps ultimately distinguishing different 'trajectories' cells take as they age, would be improved with new methods that enable rapid purification of large quantities of replicatively aged cell populations.

Here, we introduce a robust methodology, utilizing miniature chemostats (*Miller et al., 2013*), for purifying populations of aged yeast cells that is both easily scalable and highly flexible with respect to environmental condition and strain background. With this approach, the fluidic setup is straightforward, no genetic engineering of strains is required, and the cells are aged in a near-constant environment with fresh media continuously provided. Applying next-generation sequencing-based approaches, we profile chromatin and transcriptome changes (using ATAC-seq and RNA-seq) in replicatively aging yeast cells. Comparing our data to the literature, we were able to verify some observations of aging cells, challenge a prominent dogma, and identify several features specific to aging cells. Consistent with previous findings, we observe a clear connection between aging, slow growth, and stress. Contrary to previous findings, we find evidence that a global increase in DNA accessibility is not necessarily a general feature of aged cells. We explore the relationship between accessibility and instability at the rDNA locus and find a novel connection between proteasome activity and regulation of rDNA and nucleolar morphology with age. Collectively, we expect the genomic datasets and technological developments presented herein to accelerate systems-level studies of cellular aging in *Saccharomyces cerevisiae*.

## Results

### Miniature-chemostat aging devices (MADs) enable the enrichment of large numbers of replicatively aged yeast cells

To collect pure populations of aged mother cells, we first labeled the cell walls of exponentially growing cells with biotin and grew them ~12 hr overnight in liquid cultures before capturing biotinylated cells with magnetic streptavidin beads, as previously described (*Park et al., 2002*) (*Figure 1A*). Pre-growth in liquid culture after the biotin labeling improved the uniformity of the number of beads per cell, as compared to cells that were conjugated to beads immediately following biotin labeling. Following bead binding, the labeled mothers were loaded into our modified ministat culture vessel (~30–40 mL volume; see *Supplementary file 1* and *Figure 1—source data 1* for details) surrounded by custom-sized neodymium ring magnets (*Figure 1A*). Ring magnets are an important component of the system because they enable an even distribution of magnetically labeled mother cells along the glass walls of the MAD (i.e. they help minimize clumping). Peristaltic pumps were employed to provide aging cells with fresh media flow and to wash away progeny. The optical density ($OD_{600}$) of the culture media within the MAD was kept at less than 0.15 to ensure that pH and glucose remained constant in the vessels (data not shown). Mother cells were harvested by first removing the MAD culture vessels from the magnets, washing cells in growth media to remove contaminating daughter cells, and purifying mother cells with a large magnet (*Figure 1—figure supplements 1– 6*). We verified that purified mothers had aged at the expected ~2 hr division rate in YNB glucose minimal media by counting the number of replication bud scars for both haploid and diploid strains (38 hr: haploid = 15.5, diploid = 17.6; expected ~19) (*Figure 1B and C*).

Previous studies have used mother cell viability over time as a method for approximating differences in strain replicative lifespan (RLS) (*Lindstrom and Gottschling, 2009*). To test if we could

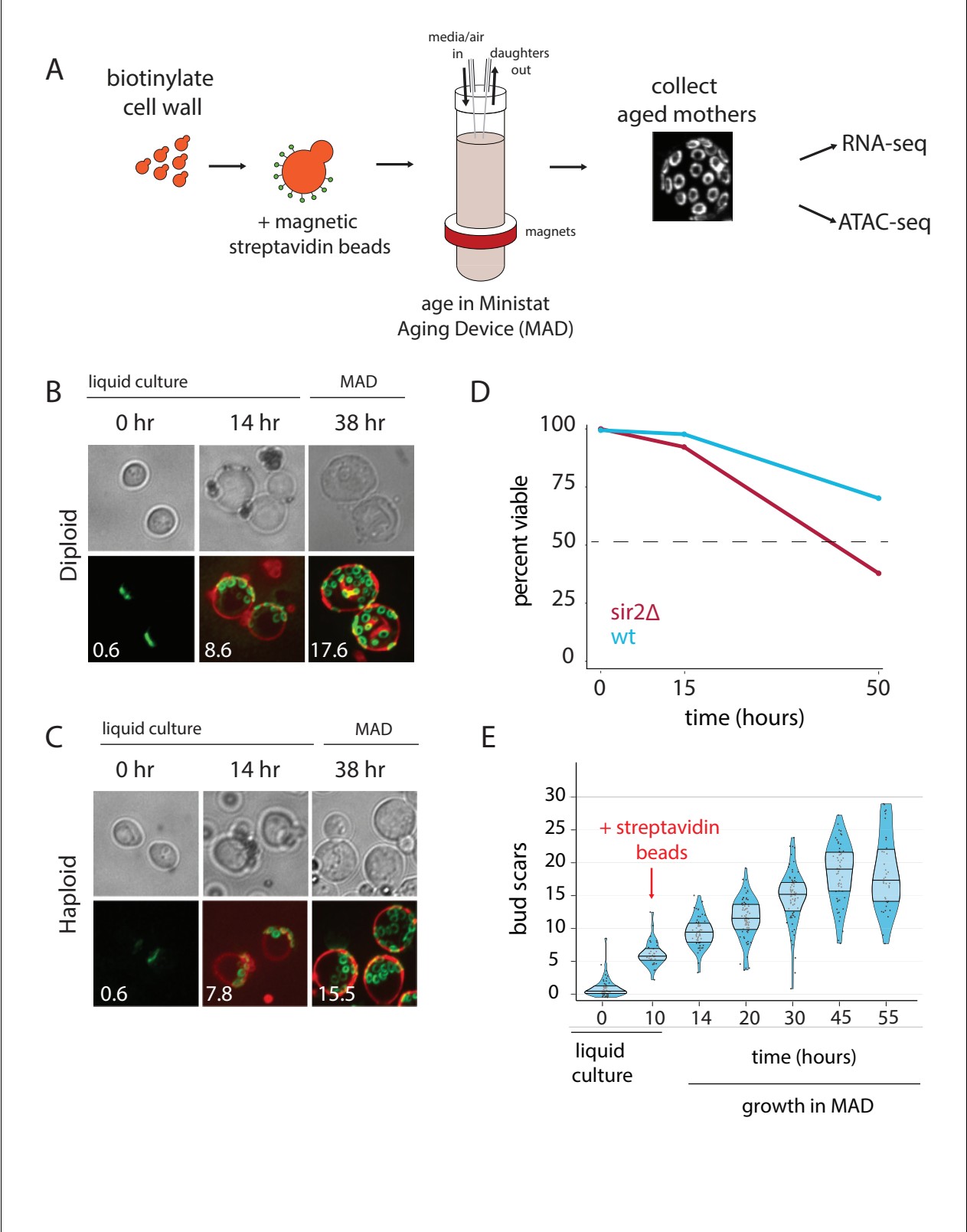

**Figure 1.** Overview of the MAD platform. (**A**) Yeast cells are biotinylated, grown in flask culture, labeled with streptavidin beads, and loaded into MAD. Daughters do not inherit the magnetic beads and are perfused from the culture, leaving behind a pure replicatively aged population of cells. (**B,C**). Haploid (DBY12000) and diploid (DBY12007) cells purified using MAD. Bud scars are labeled in green with wheat germ agglutinin conjugated to Alexa

*Figure 1 continued on next page*

*Figure 1 continued*

Fluor 488. Cell walls are labeled in red with NeutrAvidin conjugated to DyLight 593. (D) Viability of WT and *sir2Δ* cells from a 50 hr aging time course. (E) Mother cell ages (bud scars) during a 55 hr MAD time course.

DOI: https://doi.org/10.7554/eLife.39911.002

The following source data and figure supplements are available for figure 1:

**Source data 1.** List of MAD components with ordering information.

DOI: https://doi.org/10.7554/eLife.39911.009

**Source data 2.** Physiological parameters collected from dense aging time course.

DOI: https://doi.org/10.7554/eLife.39911.010

**Source data 3.** Yeast transcript types.

DOI: https://doi.org/10.7554/eLife.39911.011

**Figure supplement 1.** Images of the MAD platform.

DOI: https://doi.org/10.7554/eLife.39911.003

**Figure supplement 2.** Diagram of 3D-printed magnet holders with dimensions.

DOI: https://doi.org/10.7554/eLife.39911.004

**Figure supplement 3.** Diagram of spacer elements with dimensions.

DOI: https://doi.org/10.7554/eLife.39911.005

**Figure supplement 4.** Schematic of cap used for a single MAD.

DOI: https://doi.org/10.7554/eLife.39911.006

**Figure supplement 5.** Schematic of the 'bubble cage' used for preventing air bubbles from dislodging mother cells from the glass wall.

DOI: https://doi.org/10.7554/eLife.39911.007

**Figure supplement 6.** Average bud-scar counts of diploid cells harvested from MAD at different bead-to-cell ratios.

DOI: https://doi.org/10.7554/eLife.39911.008

observe differential aging of strains, we loaded separate MADs with wild type (WT) cells or short-lived *sir2Δ* cells, and measured membrane integrity as a proxy for viability with propidium iodide (PI) staining at three time points (*Figure 1D*) (*Defossez et al., 1999*). The deletion of the *SIR2* NAD-dependent histone deacetylase shortens lifespan by promoting hyper-recombination at the rDNA locus (*Kaeberlein et al., 1999*; *Gottlieb and Esposito, 1989*). As expected, viability for aged *sir2Δ* mothers was ~2-fold lower than that of aged WT mothers (*Figure 1D*).

## Mother cells aged in the MADs exhibit hallmarks of the aging yeast transcriptome

Substantial previous work has established that aging yeast cells share some common features across different methods and strains (*Janssens and Veenhoff, 2016*). As a starting point, we validated the MAD method by testing if we could recapitulate the salient features of the aging yeast transcriptome. We leveraged the ministat scalability to purify mothers over a dense aging time course, collecting aged cells from five separate ministats loaded from the same pre-culture (*Figure 1E*). Age, viability, and purity (% of recovered cells that are mothers) were measured with microscopy at each time point and were consistent with expectation from orthogonal methods (*Figure 1E*, *Figure 1—source data 2*) (*Janssens et al., 2015*; *Lindstrom and Gottschling, 2009*).

We then used Sleuth (an RNA-seq analysis software suite) to identify transcripts that change significantly with replicative age (*Pimentel et al., 2017*). The significances (q-values) of these were then plotted against the magnitudes of age-dependent expression changes (*Figure 2—figure supplement 1*, *Source Data 1*) (*Pimentel et al., 2017*). It is worth noting that we used an extensive catalog of *full-length* yeast transcripts (ORFs and non-coding RNAs, *Figure 1—source data 3*, [*Lardenois et al., 2011*; *van Dijk et al., 2011*; *Davis and Ares, 2006*; *Xu et al., 2011*; *Yassour et al., 2010*; *Arribere and Gilbert, 2013*; *Nagalakshmi et al., 2008*; *Johnson et al., 2011*; *Yassour et al., 2009*]), including 5' and 3' UTR regions, for mapping sequencing reads. Strikingly, some of the most upregulated transcripts were those arising from the rDNA locus, specifically from the Sir2p-silenced non-transcribed regions (NTS1 and NTS2). Transcription at the non-transcribed sequence (NTS) locus is an aberrant Pol II event (almost undetectable in log phase culture) that has been previously shown to increase with age and ERC accumulation (*Kobayashi and Ganley, 2005*; *Pal et al., 2018*; *Saka et al., 2013*).

Next, we analyzed gene sets that are differentially expressed as a function of increasing age for enrichment of a variety of functional annotations using the Database for Annotation, Visualization and Integrated Discovery (DAVID; [*Dennis et al., 2003*]). We found numerous significantly enriched terms related to metabolism, oxidoreductase activity, stress, and mitochondrial proteins amongst the genes whose expression increases with age (*Source Data 1*). Genes downregulated with age were enriched for multiple terms, including ribosome biogenesis, rRNA processing and translation as well as a host of other terms related to general cellular anabolic function (tRNA synthesis, amino acid biosynthesis, mRNA transport, exosome activity) (*Source Data 1*). In general, these results are consistent with previous reports that the aging transcriptome shares a high degree of similarity to that of induction of the environmental stress response (ESR) (*Janssens et al., 2015*; *Hu et al., 2014*; *Lesur and Campbell, 2004*; *Yiu et al., 2008*; *Kamei et al., 2014*). The ESR is the simultaneous activation of a set of stress-responsive pathways coincident with the downregulation of ribosomal biogenesis and it is inextricably tied to growth rate (i.e. if a yeast culture is growing slowly, the ESR is upregulated) (*Brauer et al., 2008*). Likewise, we also observed a shift away from glycolysis toward gluconeogenesis, similar to previous reports (*Lin et al., 2001*; *Koc et al., 2004*). Thus, MAD robustly captures the transcriptomic hallmarks of yeast aging that have been reported across a diversity of strains and methodologies (*Janssens and Veenhoff, 2016*).

## The ESR is a robust feature of the aging transcriptome

We attempted to resolve the earliest age-dependent transcript changes using hierarchical clustering of genes with significant age-dependent expression (q < 0.05) (*Figure 2*). We found that age-dependent transcripts display a clear temporal progression as we observed clusters composed of early-, middle-, and late-age changes. To understand what processes and events these clusters might be representative of, we included transcript annotation categories alongside the heatmap for multiple classes of non-coding transcripts, sub-telomeric genes, TY retrotransposon genes, and ESR-induced and ESR-repressed genes (*Figure 2*).

We found that a feature of the aging transcriptome is the activation of the ESR (i.e., almost all ESR-induced genes were upregulated with age and almost all ESR-repressed genes were downregulated [*Figure 2*]). This signature, even at later time points (where the purity of mother cells decreases), cannot be explained by newly produced daughter cells (*Figure 2—figure supplement 2A*). We also observed an age-independent initial rise and decline prior to the age-dependent increase in ESR induction in response to the beading process at the 10 hr time point. Subsequent iterations of the protocol were modified to reduce beading stress (Appendix 1); age-dependent changes and ESR induction from both protocols were highly correlated confirming that our observations are reflective of yeast aging in general.

In order to extend our comparison between the ESR and the aging transcriptome, we used Growth Rate Slopes (GRS) from *Brauer et al. (2008)*. By modulating the growth rate using nutrient limitation and measuring global patterns of gene expression across a range of growth rate, Brauer *et. al.* characterized the magnitude/direction of the effect of growth rate on gene expression as the *slope* of the regression of gene expression on growth rate (*Brauer et al., 2008*). Gene-specific GRSs have a straightforward biological interpretation. Genes with negative GRSs increase expression as growth rate slows, while genes with positive GRSs decrease expression as growth slows. We plotted the moving average for 100 gene bins across the heatmap in *Figure 2* and found that the relationship between age-related change and stress-related change extends to almost all genes for which a GRS was measured.

## Dense profiling of aging yeast cells reveals early transcriptomic events independent of the ESR

Our data and the literature agree that the aging yeast cell experiences stress/slowed growth (*Zhang et al., 2012*); however, the root cause[s] remain elusive. We reasoned that potentially causative events would occur both before and independently from the stress signature. Thus, we highlighted ESR-independent transcripts and processes that happen earliest in our time course as prospective initiating agents of yeast replicative aging. As such, we isolated and analyzed age-induced gene clusters that were relatively depleted of ESR-induced genes (*Figure 2* red bars, *Source Data 1*). Strikingly, this group of genes was highly enriched for TY transposon genes

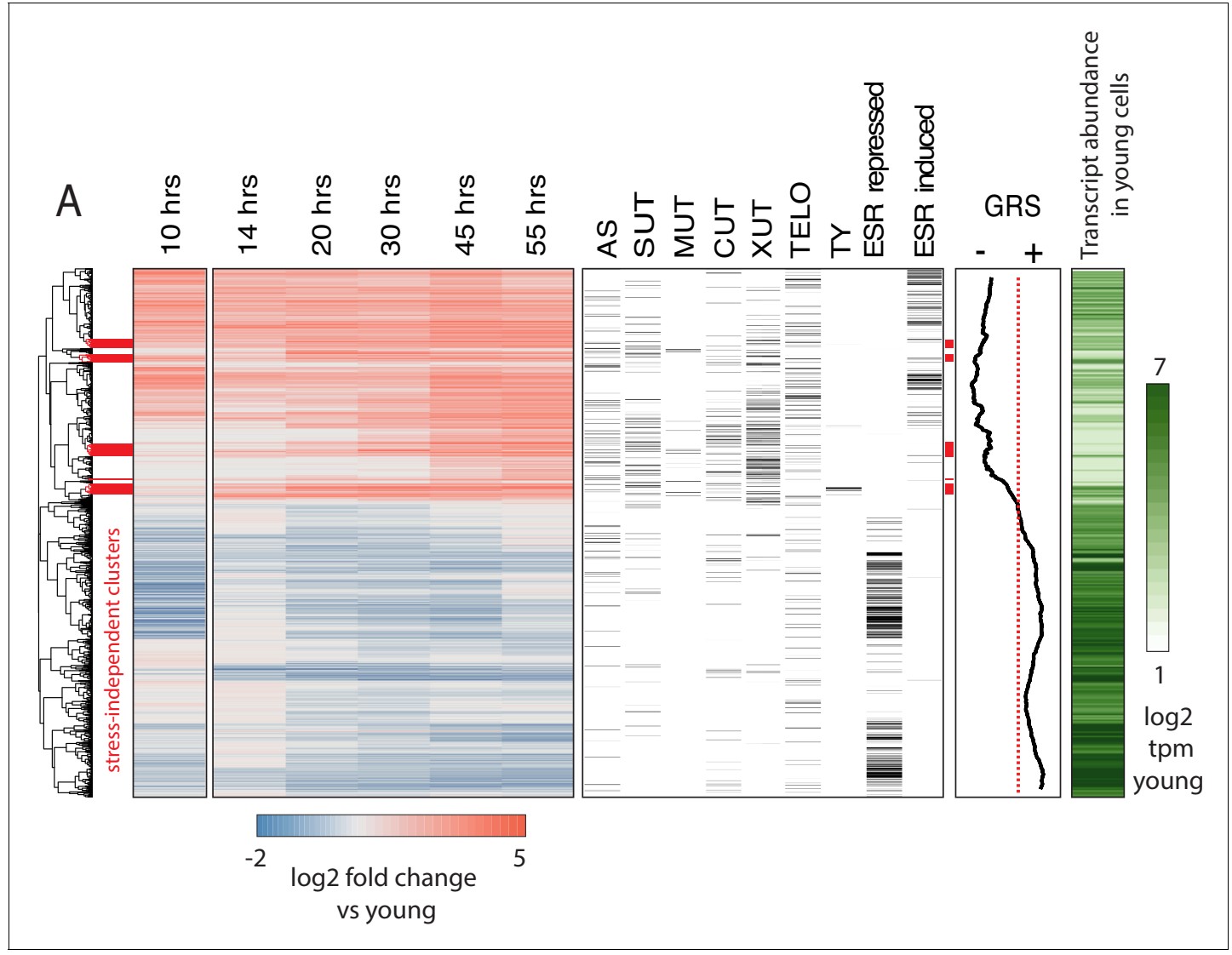

**Figure 2.** Age-dependent gene expression dynamics. (left) Hierarchically clustered heatmap of transcriptome at increasing replicative ages of ~ 2900 genes (the transcriptome at 10 hr is conflated with cell beading). Different transcript types are labeled in black directly to the right of the transcriptional data. (AS = antisense, SUT = stable unannotated transcript, MUT = meiotic unannotated transcript, CUT = cryptic unstable transcript, XUT = Xrn1 sensitive unstable transcript, TELO = subtelomeric, TY = TY repeat element, ESR = environmental stress response). (right) Moving window average (of 100 gene bins) of Growth Rate Slopes from *Brauer et al.* and log2 transcript abundances in transcripts per million during initial log-phase growth (green). Red bars indicate clusters displaying early-age induction independent of the ESR.

DOI: https://doi.org/10.7554/eLife.39911.012

The following source data and figure supplements are available for figure 2:

**Source data 1.** Physiological parameters collected from aging time courses.
DOI: https://doi.org/10.7554/eLife.39911.016

**Source data 2.** Strains.
DOI: https://doi.org/10.7554/eLife.39911.017

**Figure supplement 1.** Enriched GO terms in the aging transcriptome.
DOI: https://doi.org/10.7554/eLife.39911.013

**Figure supplement 2.** Further characterization of the aging transcriptome.
DOI: https://doi.org/10.7554/eLife.39911.014

**Figure supplement 3.** Batch effects drive the age-dependent global expression increase observed in *Hu et al., 2014*
DOI: https://doi.org/10.7554/eLife.39911.015

(*Figure 2*, *Source Data 1*). Although age-dependent increases in transposon expression have been previously reported (*Patterson et al., 2015*), here we show that it is one of the earliest and strongest age-related changes (*Figure 2*, *Source Data 1*). We also found multiple transcripts arising from the non-transcribed spacer region (NTS) of the rDNA repeat enriched within this gene set as well as genes known to be involved with response to replicative stress and DNA damage (*RNR3*, *IRC4*) raising the possibility that rDNA instability and extrachromosomal rDNA circle (ERC) formation are very early events in the yeast aging process.

The carbon source-responsive zinc-finger transcription factor, *ADR1*, was among these genes and was previously reported to be an early-age transcriptional responder (*Kamei et al., 2014*). *ADR1* has also previously found to be involved in regulating gluconeogenesis during the diauxic shift, raising the possibility that *ADR1* has a role in regulating the age induced transcriptional shift towards gluconeogenesis (*Wierman et al., 2017*). In addition, these clusters contained regulators of oxidative stress that are induced with age, including *SRX1*, *SOD2*, *TSA2*, *GPX2* and *XBP1*. Together, these data revealed several compelling leads that are independent of the ESR for further study into the originating deleterious events during replicative aging in yeast.

## Low abundance ORFs (at young ages) and non-coding RNAs tend to increase with age

In general, we observed a negative correlation between transcript abundance during early age and age-related induction (i.e. low abundance transcripts tend to undergo the largest fold increases; *Figure 2—figure supplement 2B*). Interestingly, we also found that non-coding transcripts (*Figure 2*, right panel) typically fell into the age-induced clusters, similar to previous results (*Sen et al., 2015*). A recent study from Hu *et al.* suggested that all transcripts increase with age as a result of pervasive global transcription stemming from nucleosome loss (*Figure 2—figure supplement 3*) (*Hu et al., 2014*). Consistent with this hypothesis, we observed that many induced genes are either expressed at low or undetectable levels during exponential growth (*Figure 2*, right bar, white to green). However, a universal increase in transcription with age *cannot* explain numerous induced genes that are highly expressed and functionally related. The tendency of low abundance and silenced transcripts to increase with age might be explained by the decrease in exosome components we observed, as well as a decline in mRNA degradation pathways (*Figure 2—figure supplement 1*) and decreasing transcriptional fidelity (*Sen et al., 2015*). This is consistent with the observation that the exosome component *XRN1*-sensitive unstable transcripts (XUTs) and cryptic unstable transcripts (CUTs) tend to be age-induced (*Figure 2*, *Source Data 1*). Contrary to a previously published aging transcriptomic dataset (*Hu et al., 2014*), our data indicate that universal up-regulation of gene expression is not a general feature of aging cells. We suggest that this discrepancy may partially be due to batch effects in the previously published data (*Figure 2—figure supplement 3*).

## The aging transcriptome is highly correlated across a mutant panel

Although multiple wild-type strains have been transcriptionally profiled during the aging process, to our knowledge, the aging transcriptome of mutants has not been studied. It is unknown if long- and short-lived strains experience the same process albeit on a different timescale or if there are clear differences in what stresses RLS mutants undergo as a function of age and how their response might differ as a function of genotype. In the event of the latter scenario, perhaps different long-lived strains share aspects unique from that of wild type such that a 'protective' transcriptional response or state might be identified. To explore this idea further, we chose to profile *ubr2Δ* and *fob1Δ*, whose RLS phenotypes are reported to be independent. Deletion of *FOB1* can modulate yeast RLS by increasing rDNA stability and reducing ERC formation (*Defossez et al., 1999*; *Takeuchi et al., 2003*), whereas deleting *UBR2* may confer longevity through constitutive upregulation of proteasome activity (*Kruegel et al., 2011*). In addition, we selected the *sir2Δ* strain to profile with age as well. As mentioned above, *SIR2* silences transcription of the NTS rDNA region and is a major regulator of rDNA stability.

We first determined that our mutant panel had similar growth rates (*Figure 3—figure supplement 1A*) and could be grown in the MAD for equivalent amounts of time. We then grew and labeled mother cells as described above for growth in MADs and sampled cells at three timepoints: young cells (exponential growth phase), middle-aged cells (mothers with 20 hr total growth), and old

cells (mothers 40 hr of total growth). We ran three separate aging campaigns separated by 1 week as independent biological replicates and prepared samples for RNA-seq and ATAC-seq. In all experiments, purity and viability were assessed using flow cytometry as previously reported and median age of mother populations via microscopy ([*Janssens et al., 2015*], *Figure 3—figure supplement 2*, *Figure 2—source data 1*, *Figure 2—source data 2*).

We then checked that we could observe expected differences between strains, such as up-regulation of proteasome components in the *ubr2Δ* strain (*Figure 3—figure supplement 1B*) and loss of expression of deleted genes (*Figure 3—figure supplement 1C*). Surprisingly, we also observed an apparent increase in expression of proteasome-related genes in the *sir2Δ* strain (*Figure 3—figure supplement 1B*). Importantly, we observed a significantly lower median bud scar age at both aged time points for the *sir2Δ* strain compared to wild type and slightly higher ages for *ubr2Δ* and *fob1Δ* compared to wild type in both the total population of collected cells and in the viable fraction (*Figure 3—figure supplement 3*). That cell division slows in the approach to replicative senescence is a well-documented observation (*Fehrmann et al., 2013*). Therefore, we reasoned that the older bud scar age in *fob1Δ* and *ubr2Δ* is not a function of a faster growth rate per se, but rather a result of there being a higher percentage of actively dividing cells in *fob1Δ* and *ubr2Δ* backgrounds.

After quantifying transcript levels, we used Sleuth to call significant gene expression changes as a function of age. Notably, since we calculate the magnitude of change as an exponential fit of expression change to numbers of bud scars, our 'aging slope' can be interpreted as rate of change per cell division. We then used the aging slope to determine both the strain-dependent and strain-independent changes dependent on increasing age (*Figure 3A*, *Figure 3—figure supplement 4*). Although the aging transcriptomes were similar, it was the paucity of significant differences that emphasizes how analogous the transcriptional aging profile appears across strains (*Figure 3—figure supplement 4*). Even for *sir2Δ*, the strain that exhibited the strongest strain-specific response to aging, the significant differences were in most cases a matter of magnitude per replication event (bud scar); the same genes changed, only more so, reflective either of a higher aging rate or a larger percentage of cells close to replicative senescence (*Figure 3—figure supplement 4*).

To identify functionally coherent patterns across the significant changes common to all genotypes, we used GO-term analysis to find enriched functional annotations for the set of genes that increase with age (*Figure 3A*, red) and the set of genes that decrease with age (*Figure 3A*, blue) across the tested genotypes. The enriched annotation sets for genes that increase with age across genotype mirror the terms we found for the aging wild-type strain in our previous dense time course (e.g. stress, membrane component, oxidation-reduction processes) going up as a function of age and ribosome biogenesis and rRNA processing going down with age. Notably, we observed an age-dependent expression decrease in five out of six components of the signal recognition particle (SRP).

We compared the rate of change with age (*Figure 3B*, y-axis) to the rate of change with growth rate (*Figure 3B*, x-axis) and found a strong correlation for genes that increase with age and canonical ESR induced genes (r = −0.67). In general, the correlation for all genes (r = −0.48) was strong and underscores the close relationship between RLS, stress, and growth rate.

## Age-dependent changes in gene expression that are not explained by ESR/GRS are highly enriched for SRP targets

As with the first WT time course, we sought to extricate age-dependent changes in gene expression that are not dependent on growth rate and the ESR. Accordingly, we focused on the set of significant age-related changes bounded by a GRS between −5 and 5, a regime wherein the age-dependent changes in gene expression that we observe are not clearly dominated by the apparent relationship between age-related change and GRS/ESR (*Figure 3C*). We next asked whether this class of genes was a representative subset of all age-dependent changes genes or if it was enriched for specific functional annotation sets using GO term analysis (*Source Data 2*). We found that genes with an age-related increase in expression that are not strongly dependent on the GRS are greatly enriched for genes coding for transmembrane helices and more specifically for those that are classified as secreted and/or of containing sequence for a signal recognition peptide (SRP binding site) (*Source Data 2*). Next, we compiled a more exhaustive curated list of transcripts predicted or known to associate with the SRP (signal sequence-containing and SRP-binding; [*del Alamo et al., 2011*], *Source Data 2*, *Figure 3C* green dots). We found that these SRP clients are, as a class of genes,

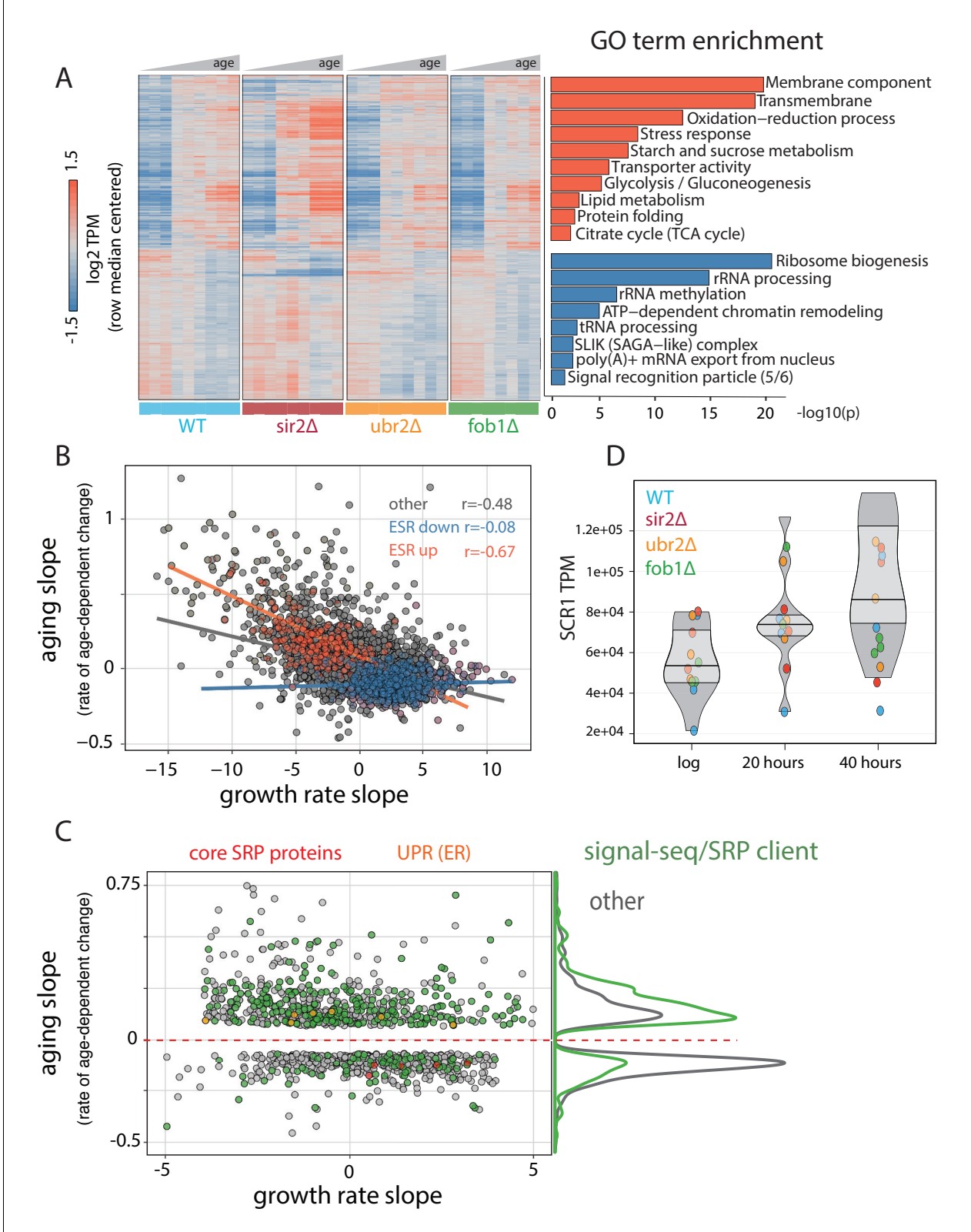

**Figure 3.** Common features of the aging transcriptome across multiple genotypes. (**A**) (left) Hierarchically clustered heatmap of ~1500 genes with significant age-dependent transcriptional responses across multiple genotypes. (right) Significance of GO terms in upregulated and downregulated genes, respectively (-log10(p)). Experiments were performed in triplicate. (**B**) Scatterplot of 'aging slopes' from (**A**) versus 'growth rate slopes' from Brauer *et al.* Canonical ESR-induced transcripts are labeled red and canonical ESR-repressed transcripts are labeled blue. (**C**) Subset of (**B**), filtered for

*Figure 3 continued on next page*

*Figure 3 continued*

genes with aging slopes q < 0.05 and growth rate slopes > 4 or < −4 removed. SRP-encoding genes are labeled red, UPR genes are labeled orange, and SRP clients are labeled green. Right panel: density of SRP clients (green) and other genes (grey) across significant aging slopes. (**D**) Gene expression levels of *SCR1* at increasing ages.

DOI: https://doi.org/10.7554/eLife.39911.018

The following figure supplements are available for figure 3:

**Figure supplement 1.** Characterization of mutant panel.

DOI: https://doi.org/10.7554/eLife.39911.019

**Figure supplement 2.** Schema for flow cytometric analysis of aged yeast cells.

DOI: https://doi.org/10.7554/eLife.39911.020

**Figure supplement 3.** Bud scar counts for MAD collected aged mutants.

DOI: https://doi.org/10.7554/eLife.39911.021

**Figure supplement 4.** Strain specific changes in the aging transcriptome of mutant panel.

DOI: https://doi.org/10.7554/eLife.39911.022

**Figure supplement 5.** SRP clients increase in expression as a function of decreasing SRP components.

DOI: https://doi.org/10.7554/eLife.39911.023

**Figure supplement 6.** *SCR1* expression increases with age in the dense WT time course.

DOI: https://doi.org/10.7554/eLife.39911.024

**Figure supplement 7.** Expression of telomere proximal genes increases with both replicative age and stress.

DOI: https://doi.org/10.7554/eLife.39911.025

significantly upregulated with age in a GRS-independent manner ($p < 10^{-51}$, hypergeometric, *Figure 3C*). As SRP clients tended to increase in expression with age in parallel with age-dependent downregulation of SRP subunits (*Figure 3C*, red dots), we asked if these two phenomena might be linked. To this end, we plotted the expression of the SRP subunits versus the expression of SRP clients across the yeast deleteome dataset, in which global gene expression was profiled across ~1500 deletion mutants ([*Kemmeren et al., 2014*], *Figure 3—figure supplement 5*). Expression of SRP subunits and SRP clients was strongly negatively correlated ($r \sim -0.5$), suggesting the existence of a negative feedback loop between SRP and its clients.

We next checked the expression of *SCR1,* the non-coding RNA subunit of the SRP complex. The aggregate trend across all of the genotypes we assayed indicates that *SCR1* is increasing roughly two-fold from ~5% to~10% of the transcriptome with age (*Figure 3D*). We also checked the abundance changes for *SCR1* in our dense time course and found the same upward trend with age (*Figure 3—figure supplement 6*). Since we observed discordant trends for the SRP protein component mRNAs and *SCR1* (non-coding RNA), as well as an increase in expression of SRP clients, we looked for a signature of ER stress in our expression data. Genes involved in ER-associated protein degradation, specifically the members of the Cdc48p-Npl4p-Ufd1p segregase complex, were upregulated with age (*Figure 3C*, orange dots) and are not canonical ESR genes.

## Proximity to telomeres correlates with age-dependent increases in gene expression

Clustering analysis revealed that genes positioned in subtelomeric regions are more likely to increase with age than to decrease (*Figure 2*). In addition, previous work has suggested a loss of silencing at subtelomeric regions, potentially as a result of redistribution of *SIR* proteins from telomeres to the nucleolus, follows from an increasing rDNA copy number and age-dependent nucleolar dysfunction that is not dependent on telomere length (*Kennedy et al., 1997*; *Salvi et al., 2013*; *Kim et al., 1996*). We thus sought to determine to what extent telomere proximity affects the expression of a gene as a function of replicative age. We looked at rate of change with age for every gene versus the distance in base pairs from the telomeres for each genotype (*Figure 3—figure supplement 7A*). We found that genes located in subtelomeric regions are more likely to increase in expression with age (*Figure 3—figure supplement 7A*). Given the link we observed between increases in gene expression with age and increased expression as a function of slow growth and stress discussed above, we next asked if the observed age-dependent upregulation of sub-telomeric gene expression was also similarly connected. We plotted the GRS versus distance from the

telomeres and found that indeed, subtelomeric genes as a group have a negative GRS and thus increase in expression as a function of slower growth rate (*Figure 3—figure supplement 7B*).

## Evidence that global nucleosomal occupancy does not decrease with age

Having characterized the aging transcriptome across a panel of strains, we sought to thoroughly characterize age-dependent changes in chromatin structure. ATAC-seq is an assay for measuring chromatin accessibility and nucleosomal occupancy and positioning by counting insertions of Tn5 transposase in the genome (*Buenrostro et al., 2013*). We used ATAC-seq to examine changes in the chromatin structure during yeast aging. We extracted transposase insertions from end points of ATAC-seq fragments and counted the number of insertions at each genomic location as a proxy of chromatin accessibility. To correct for variable sequencing depth and different amounts of rDNA between samples, insertion counts were normalized so that the total number of non-ribosomal, non-mitochondrial, uniquely mapped insertions was the same between samples. Precision of high-resolution analyses of chromatin structure is sensitive to read depth (*Schep et al., 2015*), and therefore we sequenced each sample deeply, requiring at least 25 million uniquely mappable ATAC-seq fragments per sample.

It was recently suggested that old cells have a globally lower nucleosomal occupancy that could result in permissive transcription and genomic instability (*Hu et al., 2014*). Since that study was limited to very old cells, we used ATAC-seq data to look for changes in nucleosomal occupancy as a function of time and in different genetic backgrounds. Nucleosomes protect DNA from transposition by Tn5, resulting in depletion of Tn5 insertions in a region of about 140 bases around the centers of nucleosomes (*Figure 4A*) (*Brogaard et al., 2012*). These regions became more accessible in a sample from the old cells, consistent with global loss of nucleosomes. We noticed, however, that ATAC-seq samples from the old cells contained a higher level of background insertions. For instance, mid-gene bodies that are normally inaccessible to ATAC-seq became uniformly more accessible (*Figure 4B*). This led us to examine an alternative hypothesis, that the non-specific background signal was uniformly higher in the samples from the old populations, leading to an apparent gain of accessibility in nucleosomal regions. Consistent with this hypothesis, the accessibility of canonically open nucleosome-free regions (NFRs) decreased with age (*Figure 4—figure supplement 1A*). Indeed, since our data was normalized in a way that every sample has the same number of ATAC-seq insertions, increased insertions in the closed regions 'compete' with the insertions in the open regions, resulting in an appearance of decreasing accessibility of the open regions (*Figure 4—figure supplement 1A*).

What can increase the background? Naked DNA from dead cells has been a prominent confounder in microbiological studies as extracellular DNA can remain stable for days (*Nielsen et al., 2007*). DNA from the dead cells loses its chromatin structure, yielding an ATAC-seq signal that is virtually uniform (*Figure 4—figure supplement 2*). Because the number of dead cells recovered (*Figure 1D*) increases with age, we hypothesized that the non-specific background in the ATAC-seq signal could rise because of the contamination by dead cells in old cultures. Since dead cells lose their membrane integrity, their DNA can be made inert to the ATAC-seq amplification by pre-treatment with intercalating agent propidium monoazide (PMA). PMA selectively enters dead cells and upon photo-activation binds covalently to DNA, strongly inhibiting its amplification in subsequent PCR reactions. Testing the efficacy of PMA treatment in yeast on heat-killed cells revealed that PMA can effectively remove the 'dead cell' signature from mixed samples (*Figure 4—figure supplement 2*). We, therefore, compared ATAC-seq signal with and without pre-treatment of the sample with PMA.

Strikingly, addition of PMA decreased the accessibility of nucleosomal DNA in the old sample to the level observed in the young sample (*Figure 4A*), and this PMA-dependent decrease in accessibility was specific to the old cells (>30 hr) (*Figure 4—figure supplement 1B*). To verify that the average effect does not come from few exceptionally accessible nucleosomes, we binned nucleosomes into five accessibility bins and observed the same effect (*Figure 4—figure supplement 1C*). Further, chromatin structure around promoters became virtually indistinguishable from that of the young cells and the non-specific background in the gene bodies decreased to the level of the young cells, consistent with a significant presence of the DNA from the dead cells in the old-cell samples.

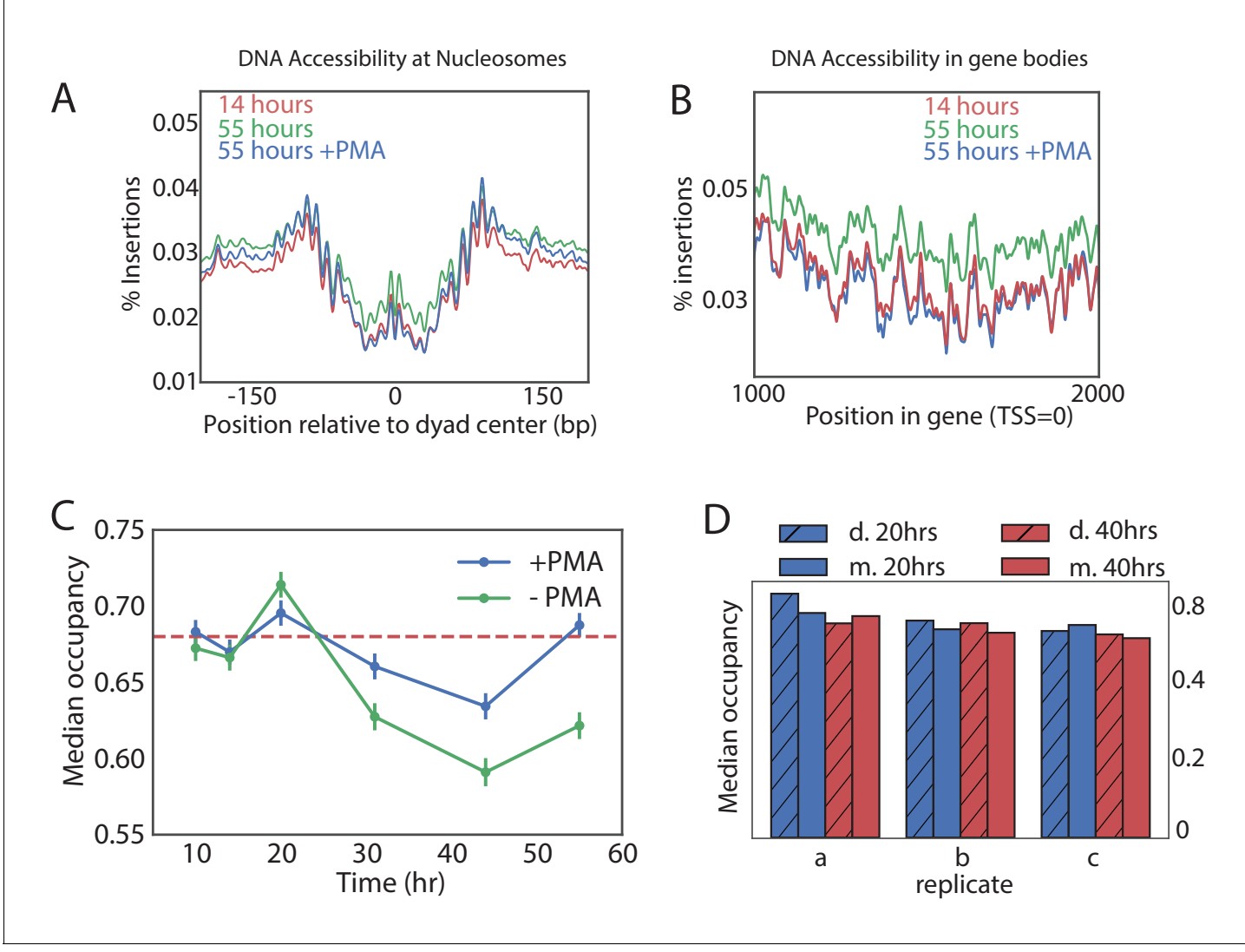

**Figure 4.** Global nucleosome occupancy remains nearly constant with aging. (**A,B**) Tn5 insertion density around well-positioned nucleosomes (**A**) and 1000 bases downstream of TSS (**B**) at 14 hr, 55 hr and 55 hr after treatment with PMA. (**C**) Median nucleosomal occupancy estimated by NucleoATAC on well-positioned nucleosomes in open chromatin at different time points during time course. See Materials and methods for the method of estimation. Error bars represent five standard errors of occupancy. (**D**) Same as (**C**) but comparing mothers and daughters during aging time courses - three independent replicates are shown.

DOI: https://doi.org/10.7554/eLife.39911.026

The following figure supplements are available for figure 4:

**Figure supplement 1.** PMA treatment rescues young cell ATAC-seq profile at transcriptional start sites in old cell populations.
DOI: https://doi.org/10.7554/eLife.39911.027
**Figure supplement 2.** Efficacy of PMA in removing the ATAC-seq signature of heat-killed cells from mixed populations.
DOI: https://doi.org/10.7554/eLife.39911.028
**Figure supplement 3.** Mean nucleosomal occupancy estimated by NucleoATAC on well-positioned nucleosomes in open chromatin.
DOI: https://doi.org/10.7554/eLife.39911.029
**Figure supplement 4.** ATAC-seq analysis.
DOI: https://doi.org/10.7554/eLife.39911.030

To quantitatively measure the effect of age on nucleosomal occupancy, we used NucleoATAC to measure nucleosomal occupancy around the well-positioned nucleosomes in areas of open chromatin (*Schep et al., 2015*). Indeed, without PMA treatment, average nucleosomal occupancy declined with age. This decline in occupancy, however, nearly disappeared when the samples were treated

with PMA (*Figure 4C*). The effect of PMA was specific to the old cells, consistent with the increase in the dead cell proportion causing a seemingly declining occupancy (*Figure 4C*, *Figure 4—figure supplement 3*). These findings were further supported by comparing occupancy of the PMA-treated daughter cell fraction (flow-through) to that of the middle aged and old mother cell fractions in three biological replicates (*Figure 4D*). Also here, we observed that the nucleosomal occupancy remains constant, on average, with age (*Figure 4D*).

We conclude that the global chromatin structure does not deteriorate significantly with age and previous observations to the contrary may be explained by an increase in the proportion of the dead cells in the aging population.

## Age-dependent changes in chromatin accessibility

Although global chromatin structure did not change significantly between the young and the old cells, local changes in chromatin structure with age were observed (*Figure 4—figure supplement 4A*), suggesting that the activities of multiple transcription factors and chromatin remodelers change with age. To define age-dependent opening and closing of genomic regions, we divided the yeast genome into 100 bp long non-overlapping bins and performed a statistical test of age-dependent opening and closing (see Materials and methods).

We first looked for common age-dependent chromatin changes between the four mutants. In general, at any significance cutoff more bins were closing than opening. We set the significance cutoff at a q-value of $10^{-3}$. At this q-value approximately 12% of bins were significantly opening or closing in all strains (*Figure 5A*, *Figure 4—figure supplement 4B*, *Source Data 3*). Bins located in gene bodies tended to preferentially open, while promoters tended to preferentially close. Origins of replications (ARS) strongly tended to lose accessibility (*Figure 5A,B*, *Figure 4—figure supplement 4C*, *Figure 4—figure supplement 4D*). Comparing with other ATAC-seq datasets in yeast, we found that a similar closing of ARS elements occurs during the transition from the reductive to oxidative phase of the metabolic cycle (*Gowans et al., 2018*) (*Figure 4—figure supplement 4E*). Since the reductive phase is characterized by a global decrease in DNA replication, our results suggests that DNA replication is inhibited in aging mothers.

We then asked if changes in chromatin accessibility reflect changes in activity of certain chromatin regulators or transcription factors through aging. We assigned each changing ATAC-seq bin to the

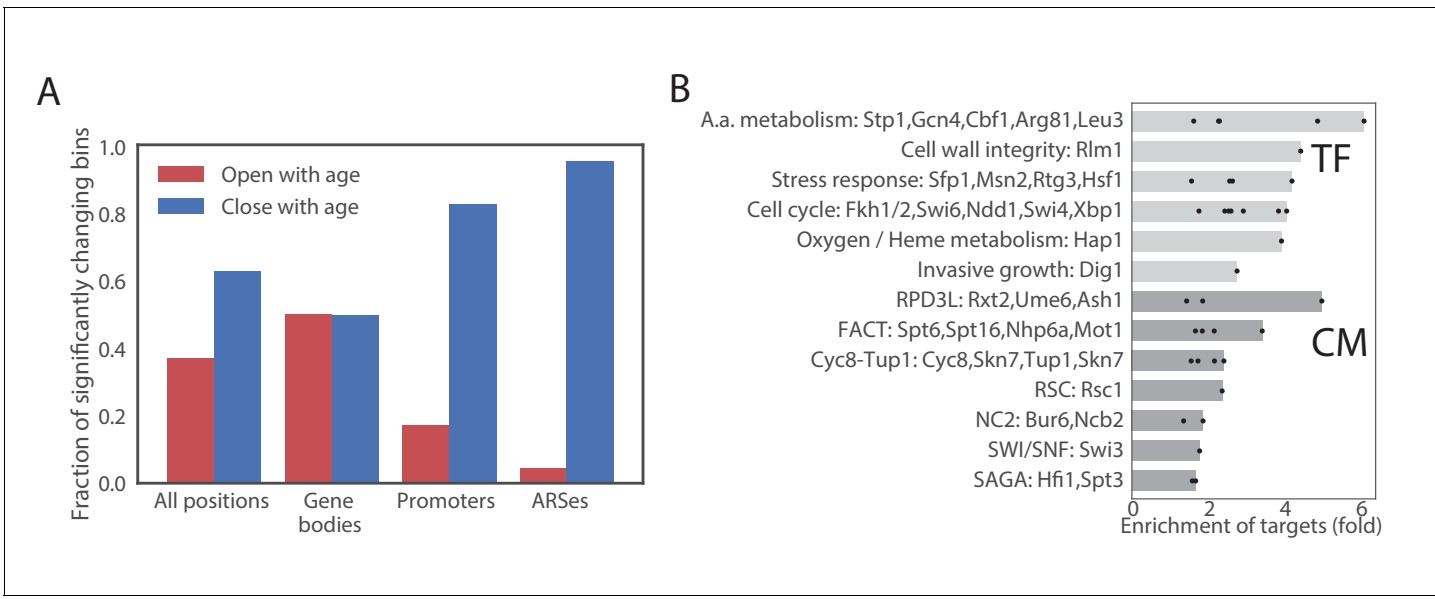

**Figure 5.** Age-dependent changes in chromatin accessibility. (**A**) Fraction of significantly changing bins that are opening or closing with age in each part of a genome. (**B**) Enrichments of genes regulated by particular transcription factors (TF) and chromatin modifiers (CM) among genes that contain bins that open with age. TFs/CMs are ordered in the order of increasing fold-enrichment and the bar marks the maximal fold-enrichment of a factor in the group.

DOI: https://doi.org/10.7554/eLife.39911.031

closest gene and queried published datasets (*MacIsaac et al., 2006*; *Venters et al., 2011*) to determine if genes that contain changing bins are bound by a particular factor. No strong enrichment was found in bins that decrease in accessibility, but bins that increase in accessibility were highly enriched for binding of multiple transcription factors, including cell cycle regulators (Ndd1, Swi4/6, Fkh1/2 and Sbp1), and regulators of amino acid biogenesis, particularly Arg81, Leu3 and Gcn4 (*Figure 5B*, *Source Data 4*). Notably, all enzymes in leucine biosynthesis superpathway and all enzymes in arginine biosynthesis pathway besides Arg2 and Arg7 gained accessibility with age.

To test if the age-dependency in chromatin accessibility differs between the mutants, we plotted the estimated age dependent slopes of bins in different genetic backgrounds. The aging slopes estimated in WT, *ubr2Δ* and *fob1Δ* were highly correlated with a slope of one. *Sir2Δ* cells, however, behaved quite differently, with all slopes significantly higher than in the wild type, which we interpret as faster rate of aging in this background. Notably, however, although the slopes of changes were higher in *sir2Δ* overall, the slopes were highly correlated, suggesting that the overall pattern of age-dependent change in accessibility remains the same in this mutant strain (*Figure 4—figure supplement 4F*).

## Accessibility and relative copy number at the rDNA locus is genotype dependent

Despite the similarity in aging ATAC-seq profiles across our mutant panel, we hypothesized that accessibility at the rDNA locus would be an important distinguishing feature as *SIR2* and *FOB1* are known modulators of rDNA stability. We reasoned that ATAC-seq insertional density at the rDNA locus would report on three distinct outcomes with respect to increasing instability: (1) increased rDNA accessibility, (2) increased rDNA copy number (extrachromosomal), and (3) increased rDNA copy number (genomic). We found that with age, WT strains present with a ~ 2 fold increase in insertional density over the entire rDNA locus (*Figure 6*, first panel). To determine whether the observed increase in ATAC-seq coverage at the rDNA locus (*Figure 6A*) was driven by increased accessibility (1) versus copy number expansion (2 and 3), we quantified relative rDNA copy number using qPCR with DNA from aged samples, and found that rDNA copy number increased with age (*Figure 6B*). In fact, in comparison to insertional density, relative copy number increases to a larger extent with age, *suggesting* that the age-dependent increase in insertional density at the rDNA locus is largely being driven by copy number expansion. We observed that the increased ATAC-seq signal at the rDNA locus returned to normal levels in daughters born to old mothers, demonstrating an asymmetric division of signal. Together, these data argue for (3): increasing ATAC-seq insertion density at the rDNA locus is reflective of greater numbers of asymmetrically retained ERC rDNA repeats with age.

We looked at rDNA ATAC-seq coverage in *fob1Δ* and *sir2Δ* mutants, which repress and promote ERC formation, respectively. *Sir2Δ* cells had slightly more (~20%) relative rDNA copy number than WT at young ages (*Figure 6A,B*) but much higher ATAC-seq coverage (twofold). However, in looking at age-related increases at this locus in *sir2Δ*, relative rDNA copy number increased by ~10 fold and ATAC-seq signal only increased by twofold at older ages; one interpretation of these data is that an initial open chromatin state may promote instability/higher copy number with age.

Surprisingly, like the *sir2Δ* strain, *fob1Δ* cells begin with ~15% higher relative rDNA copy number compared to WT during log phase, and have actually ~20% higher accessibility as measured by rDNA ATAC-seq than WT, consistent with the role of Fob1p in Sir2p recruitment to the rDNA (*Huang and Moazed, 2003*). We find that *fob1Δ* cells experience a much more modest increase in both ATAC-seq signal (1.4 fold vs 2.5 fold) and relative copy number (1.6-fold vs 6-fold) with age. Importantly, we note that if the source of our increased ATAC-seq rDNA coverage and estimated relative rDNA copy number are in fact from ERC build up, *fob1Δ* is only partially defective in ERC formation, consistent with previous work (*Lindstrom et al., 2011*). In young cells, rDNA ATAC-seq signal and relative copy number for *ubr2Δ* was equivalent to WT. With age, however, the *ubr2Δ* cells displayed an attenuated rDNA phenotype compared to WT in both ATAC-seq (1.7 vs 2.5) and relative copy number (2.3 vs 6).

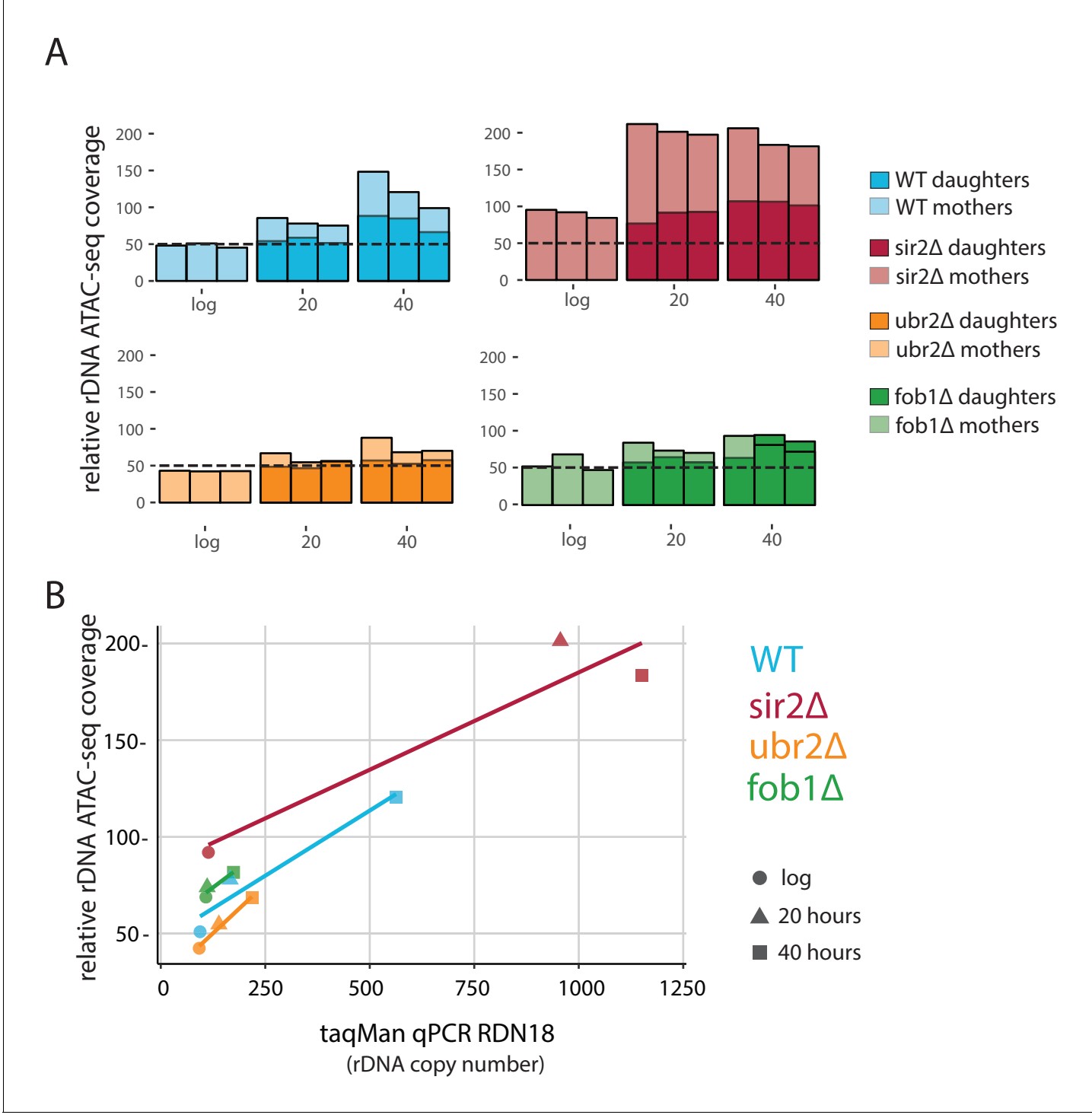

**Figure 6.** *fob1Δ* and *ubr2Δ* reduce age-dependent rDNA instability. (**A**) Percent relative (to non-repetitive genome) ATAC-seq insertional density at the rDNA locus at increasing ages in both mother and daughter cells of WT and longevity mutants. Bars are 'layered' with data from daughter cells (opaque) displayed on top of data from mother cells (semi-transparent). Three independent replicates are shown. (**B**) Comparing relative rDNA ATAC-seq insertional density (y-axis) to relative rDNA copy number using TaqMan assay (x-axis) for multiple strains.

DOI: https://doi.org/10.7554/eLife.39911.032

The following source data is available for figure 6:

**Source data 1.** TaqMan primers and amplicons.

DOI: https://doi.org/10.7554/eLife.39911.033

## *ubr2Δ* and *fob1Δ* strains exhibit resistance to nucleolar fragmentation with replicative age

In yeast cells, rDNA is localized, transcribed, processed and assembled into ribosomes within the nucleolar subcompartment of the nucleus (*Taddei and Gasser, 2012*). It has previously been shown that the nucleoli of aged yeast cells grow large and fragment (*Sinclair et al., 1997*). To compare our ATAC-seq measurements to a cell biological phenotype, we monitored nucleolar morphology for our mutants as a function of replicative age using the MAD platform. To do so, we aged cells from each strain that also harbored a fluorescent nucleolar reporter (a translational fusion of *NOP58* and mNeonGreen). Compared to young cells, old cells from all genotypes showed nucleolar enlargement (*Figure 7*). However, the observed increases were clearly more pronounced in WT and *sir2Δ* than in *ubr2Δ* and *fob1Δ*, similar to what was seen in the qPCR and ATAC-seq assays.

## Discussion

Although the study of RLS in *Saccharomyces cerevisiae* has proven an informative model for studying how genetic and environmental factors impact cellular aging, the field has not been fully able to access the 'omic' revolution. Significant technical challenges have impeded the collection of adequate numbers of viable aged cells for genomic or biochemical assays, much less aged cells across multiple environments and genotypes. The extant systematic genomic research in yeast aging has been restricted to a handful of strains under similar conditions. Here, in order to fully leverage the environmental and genetic flexibility of yeast as an RLS model, we introduce the MAD platform as an effective methodology for aging large numbers of yeast cells in a wide variety of genetic background and environments.

We found in our analysis of the transcriptional response to aging that a dominant feature is the activation of the ESR, characterized by the induction of stress response factors and a repression of ribosome biogenesis programs. A previous study provided evidence that deleting both of the primary ESR-related transcription factors, *MSN2* and *MSN4*, does not reduce RLS on rich medium (where all of the amino acids are provided); it was also found that these same transcription factors are required for lifespan extension under conditions of lowered glucose (*Medvedik et al., 2007*). Prior work has also identified a transcriptional connection between the ESR and aging (*Lesur and Campbell, 2004*). Here, we demonstrate the strength of that relationship as evidenced by the marked correlation between the rate that a gene changes with age, and the dependence of that gene's expression on growth rate. Currently, we favor a model in which the concordance we observe follows from a cascade of cellular decline that results in age-associated slowing of growth, which is consistent with reports of age-dependent reductions in single-cell budding rates measured in microfluidics devices (*Jo et al., 2015*).

To highlight potential candidates responsible for age-induced ESR, we isolated stress-independent aging signatures by selecting genes that are not annotated as canonical ESR genes and whose expression is minimally dependent on growth rate (*Figure 3*). Analysis of these genes revealed an early induction of aberrant transcription of the NTS region of the rDNA locus, transposon upregulation, an oxidative stress response, a clear increase in the expression of SRP clients, and a cell wall integrity/ER stress signature that is independent of the ESR and GRS. Notably, we found evidence for all of these signatures in our mutant panel, albeit a much-attenuated transposon induction in the *sir2Δ* strain. Determining exactly how each of these events are interdependent versus distinct, and which might be causative, will require further study, presumably at the single-cell level.

It was previously shown that sub-telomeric regions are fast-evolving and the genes therein are enriched for being related to stress, cell surface properties, and carbohydrate metabolism, all of which are enriched functional annotations in our pan-aging gene expression signature (*Brown et al., 2010*). Those authors speculated that the high evolvability of subtelomeric regions permits rapid adaptation to novel stresses (*Brown et al., 2010*). From our analysis, close proximity to a telomere also increases the likelihood of age- and stress-dependent gene induction (*Figure 3—figure supplement 7*). Whether this telomere position effect is due to a catch-all stress response that manifests with age, or perhaps an aging-specific redistribution of sirtuin proteins, remains an open question.

Histone mRNAs and proteins have been reported to decline with replicative age in yeast (*Hu et al., 2014*; *Pal et al., 2018*). This observation, in tandem with a systematic measurement of nucleosome occupancy, have accreted into a model wherein the reductions in histone protein lower

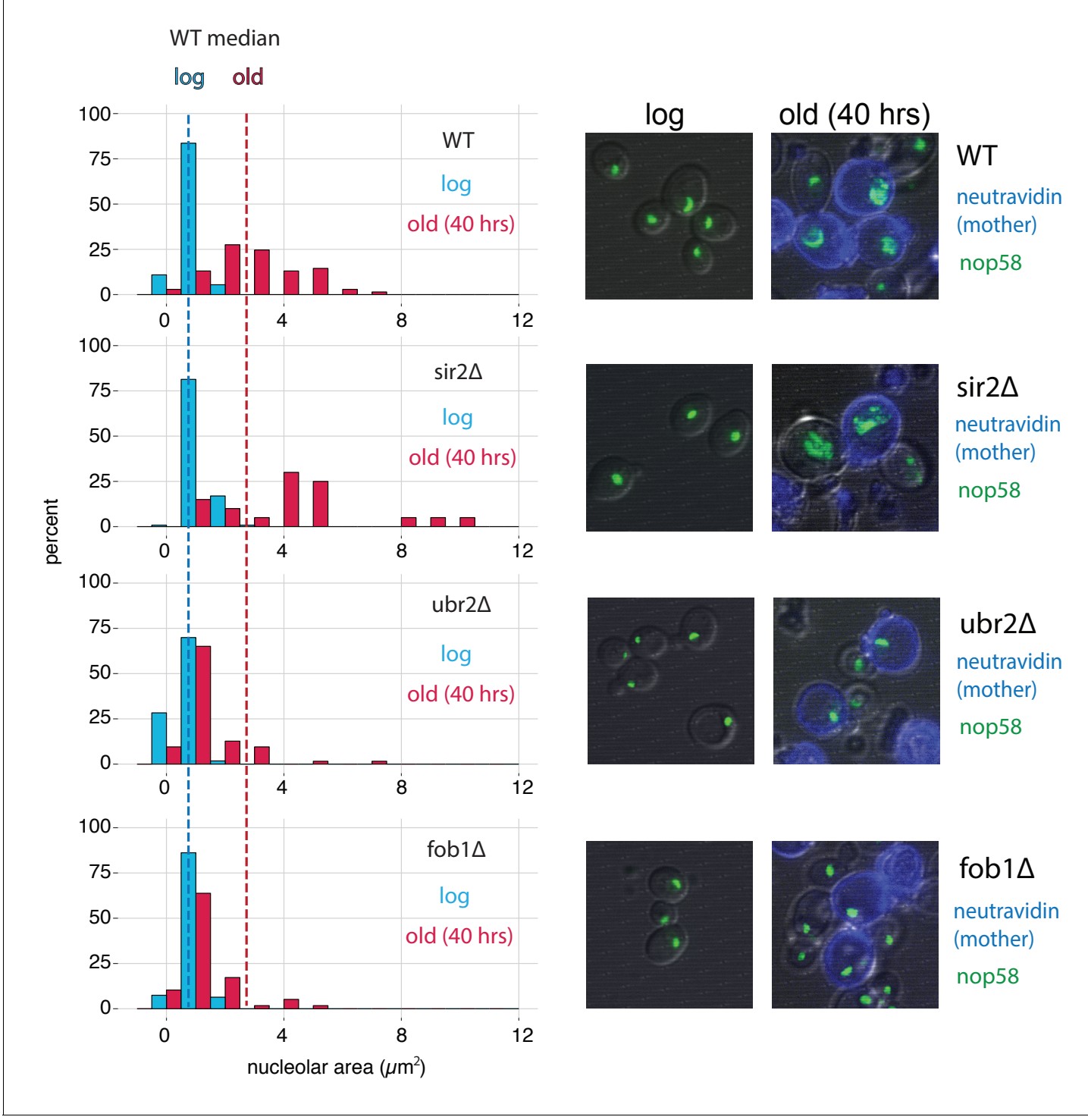

**Figure 7.** *fob1Δ* and *ubr2Δ* reduce age-dependent nucleolar fragmentation. (left) Nucleolar sizes in young (log) and old (40 hr) mother cells (displayed in blue and red, respectively) across multiple genotypes. (right) Visualization of nucleoli from young and old cells across genotypes (*fob1Δ*: $n_{old}$ = 59, $n_{young}$ = 94; *ubr2Δ*: $n_{old}$ = 64, $n_{young}$ = 115; *sir2Δ*: $n_{old}$ = 20, $n_{young}$ = 119; WT: $n_{old}$ = 70, $n_{young}$ = 55).

DOI: https://doi.org/10.7554/eLife.39911.034

the number of functional nucleosomes. The loss of nucleosomes is then purported to result in the erosion of appropriate transcriptional control which manifests as a universal upregulation of all transcripts. Combining a modified ATAC-seq protocol with MAD-purified old cells, we provide evidence that a global increase in DNA accessibility is not a general feature of aged cells. Importantly, we only observed a measurable increase in accessibility and decrease of occupancy at nucleosomes *without* the addition of the chemical agent PMA, which renders genetic input from dead cells inert. It is our expectation that our modified ATAC-seq method will become a generally useful approach for exploring genomic changes in mixed-viability samples.

We detected statistically significant changes in chromatin accessibility in ~12% of the genome. Strikingly, we found that virtually all origins of replication become less accessible as a function of age. Through comparison to other ATAC-seq datasets, we hypothesize that in an aging yeast population, a smaller fraction of cells are in S-phase and thus depleted for active replication, resulting in the observed decline in accessibility at origins of replication.

Changes in the rDNA stability with age manifest as ERC formation, which may be a common cause of replicative senescence in *S. cerevisiae* across numerous strain backgrounds and environments. Using our ATAC-seq data, we found, as expected, WT cells experience an increase in copy number of rDNA repeats that are asymmetrically retained. Likewise, we found an exaggerated and attenuated phenotype for *sir2Δ* and *fob1Δ* cells, respectively. We also found that the aged *ubr2Δ* rDNA profile phenocopies that of *fob1Δ*, an unexpected result given that *UBR2* is believed to confer longevity through proteasome activation independently of *FOB1* and *SIR2*. Overall, these results are consistent with the known roles of *SIR2* and *FOB1* and provide provocative evidence that *UBR2* is involved in ERC formation/rDNA stability.

In the future, our expectation is that combining the MAD platform with single-cell analysis could be especially informative. MAD provides the means to obtain aged cells, which can then be investigated with sequencing and optical methods; indeed, it is our expectation that single-cell RNA-seq and ATAC-seq could facilitate the identification of distinct aging trajectories where subtle causative changes can be distinguished from unsubtle stress responses (*Bendall et al., 2014*; *Trapnell et al., 2014*), making it easier to separate cause from effect. Additionally, technical improvements (such as being able to directly conjugate beads to yeast cell walls or reducing the variance in the ages of purified cells) could better facilitate detection of temporal differences between mutant strains, especially at older ages. In conclusion, by enabling the collection of multiple data modalities at different ages, genotypes, and environments, the MAD platform provides a powerful new paradigm for systems-level studies of cellular aging.

## Materials and methods

### Assembly of MADs

Individual MADs (*Figure 1—figure supplement 1A,B* are modified versions of ministats originally developed in *Miller et al., 2013*). A list of parts can be found in *Figure 1—source data 1*. There are several key components, as detailed below.

### Metro rack

A metro rack (Uline, H-2946–63) with three shelves can be used to hold 8 MADs. Casters are required to move in and out of temperature-controlled room. Shelf-liners, ledges, and push handle for safety, stability, and accessibility are optional. The top shelf is for air pumps and the bottom shelf is for effluent collection flasks. The middle shelf is for the rest of MAD components (peristaltic pump, ministat vessels in tube rack, media carboy, etc.).

### Air pumps

An individual air pump (Hydrofarm, AAPA3.2L) is used to aerate each ministat vessel. Air pressure can be adjusted with a controller to ensure optimal air flow that strong enough to provide aeration, but weak enough not to disturb labeled yeast cells attracted to magnets. Air pumps should be mounted on the metro rack with cable ties.

### Ring magnets

Ministats are threaded through three neodymium ring magnets (K&J Magnetics, RY0X04) with spacer elements (3D printed, same dimension as magnets, plastic) and placed on a rack (Fisher Scientific, 14-793-5). The ministat is then surrounded by four 3D-printed posts (*Figure 1—figure supplement 2* and *Figure 1—figure supplement 3*)

### Ministat vessel with accessories

A stainless steel (type SST316; materials from McMaster-Carr, 89495K425 and 9298K12) bubble cage (Zera Development, SBT001Rev01 and SBT002Rev01) is used to guide air bubbles to the center of the vessel so they do not dislodge the bead-labeled mother cells from the glass wall (*Figure 1—figure supplement 4*). A 7-inch needle (Hamilton, 7750–07) is connected to Y-shape branched tubing for media feeding and aeration. Two 5-inch needles (Hamilton, 7750–07) are for effluent and loading port, respectively. The loading port with male luer cap can be used to deliver drugs or other additives as well as for the inoculation of seed culture. A Teflon cap (Zera Development, SBT003Rev01; materials from McMaster-Carr, 8546K17) with a silicone gasket (Zera Development, SBT004Rev01; materials from McMaster-Carr, 1460N25) is used for positioning of the needles and airtightness (*Figure 1—figure supplement 5*).

## Cell preparation for aging in MADs

Cultures of desired strains were grown overnight at 30°C with shaking (180–200 rotations per minute [RPM]) and harvested at $OD_{600} < 0.2$ by next morning (early log phase). Upon cells reaching the desired $OD_{600}$, EZ-Link Sulfo-NHS-LC-LC-Biotin (ThermoFisher cat# 21338) was removed from −20°C freezer and equilibrated to room temperature during cell washing and preparation. For one MAD, 4 $OD_{600}$ units of cells were washed twice in PBS + 0.25% PEG3350 at 1500 g for 5 min at room temperature. During the second wash the NHS-LC-LC-Biotin was weighed out (0.5 mg per OD unit of cells) and re-suspended in 0.5 mL of PBS. Final washed cell pellets were resuspended in 0.5 mL of PBS and combined with NHS-LC-LC-Biotin in PBS for a final reaction volume of 1 mL. The labeling reaction was carried out for 30 min at room temperature with rotation followed by a 2x wash in PBS with 0.25% PEG3350, as above. Biotin-labeled cells were resuspended in growth media and grown under normal liquid culture conditions (30°C with shaking at 180–200 RPM) for 10–14 hr. At time of seeding, the number of labeled cells was determined using a Beckman Coulter Counter. In general, 4 $OD_{600}$ units of cells produces between 130 and 170 million cells with variation arising from differences in strain size and cell loss during washing. Cultures were seeded such that the cell density at time of collection would not exceed an $OD_{600}$ of 0.3.

Prior to cell collection, magnetic streptavidin beads (Dynabeads MyOne Streptavidin C1 ThermoFisher cat# 65001) were removed from 4°C storage, washed twice with 1 mL of growth media, and re-suspended in 20 mL of growth media. Following the post-biotin labeling phase of growth, cultures were passed over a 0.2 μm filter and re-suspended in 20 mL of growth media.

The number of beads used was 0.9 μL of beads per 1 million *labeled cells* (as measured previously with the Coulter counter [*e.g.* 135 μL of beads would be used for 150 million labeled cells]). Well-mixed cells from the overnight culture (original mothers + all progeny) were collected for assays (RNA-seq, ATAC-seq, etc.) as the 'log phase' sample.

Beading reactions were carried out by combining 20 mL of washed beads with 20 mL of re-suspended cells in a 50 mL falcon tube and rotating at room temperature for 15 min. Beaded mother cells were collected by placing the beading reaction on magnet (DynaMag−50 Magnet ThermoFisher Cat#12302D) for 5 min. Supernatant (daughter cells) were carefully removed and discarded. Remaining beaded mother cells were washed 2x in 40 mL of growth media and re-suspended in a final volume of 1 to 2 mL of growth media. Beading efficiency was checked and cells counted under microscope with hemocytometer. Generally, beaded cells have between 1 and 10 beads per cell with an average of ~5 and recovery is usually around 75–90% of seeded mothers. It is worth noting that keeping the bead-to-cell ratio at <20:1 is important; increasing the number of beads per cell can greatly inhibit growth (*Figure 1—figure supplement 6*). Before loading into MAD a sample of the beaded mothers was also collected for assays.

For MAD loading, ministats were removed from magnets, the air pumps were set to lowest setting, and the effluent port was pulled far above media level. Beaded mothers were loaded through

luer needle entry port using a 1000 µL micropipette. The loaded ministat was placed on magnet, and cells were allowed to bind for 10 min before restarting of pumps (50 RPM; flow rate ~25 mL/hr.) and air flow (25–33% of max output;~1 PSI). Effluent needle was lowered to ~25 mm above the top ring magnet.

For experiments in *Figures 1* and *2*, the media used was a minimal medium (Yeast Nitrogen Base [YNB] with 5 g/L Ammonium Sulfate as a nitrogen source; Fisher, DF0919) with 2% glucose and 2% mannose (mannose is used to suppress potential flocculation in the MAD). In subsequent experiments, a slightly different minimal medium was used: Yeast Nitrogen Base [YNB] with 5 g/L Ammonium Sulfate and *no* biotin (Sunrise, 1523–100) with 2% glucose, 2% mannose, and 40 nM 7,8-diaminopelargonic acid (DAPA). See Appendix 1 for more details.

## Aging of cells in MADs
Loaded cells were grown in MADs for desired length of time. For the dense time course, 5 MADs were loaded and an entire MAD worth of cells was collected per time point. For the mutant strain time courses, cells were harvested by removing the glass vessel from the magnets (pumps off, air low, and effluent needle up) and waiting 5 min for mothers to disperse. Sampling of cells was done through the loading port luer-lock syringe.

## Harvesting of daughter cells
Effluent tubing was diverted to a clean 50 mL Falcon tube to collect daughter cells. 5 million cells were spun down and frozen in liquid nitrogen for RNA, DNA preps or used immediately for ATAC-seq or bud scar counting as needed.

## Harvesting and washing of aged mother cells
Following removal of the MAD cap, tubing, and bubble cage, the supernatant was removed using a 25 mL pipette, being careful not to disturb the magnetically bound cells on the vessel wall. With the vessel removed from the magnet, beaded cells were resuspended in media and placed back on the magnet for 5–10 min. This was repeated for a minimum of four washes or until the cells were of desired purity as assessed by microscopy. Final beaded cells were resuspended in 1 mL of media and 5 million cells were spun down and frozen in liquid nitrogen for RNA, DNA preps or used immediately for ATAC-seq or bud scar counting as needed.

## Bud scar counting and viability staining
Cells were resuspended in PBS with 1M sorbitol and 4 mM EDTA and stained with 10 µL of NeutrAvidin Protein with DyLight 405 (ThermoFisher cat# 22831) or NeutrAvidin Protein with DyLight 594 (ThermoFisher cat# 22831 cat#22842) to identify biotinylated mother cells; 1 µL 5 mg/mL Wheat Germ Agglutinin, Alexa Fluor 488 Conjugate (ThermoFisher cat# W11261) to determine replicative age; and 1 µL of 5 mg/ml propidium iodide) to quantify replicative age. Cells were stained for 30 min at room temperature and then washed twice in PBS with 1M sorbitol before microscopy. Viability and purity analysis was carried out exactly as in *Janssens et al. (2015)*. FACS analysis was carried out with FloJo.

## RNA extraction
Frozen cell pellets were resuspended in 200 µL of Lysis buffer (10 mM Tris [pH 8.0], 0.5% SDS, 10 mM EDTA). Following the addition of 200 µL Acid Phenol [pH 4.3], samples were vortexed for 30 s. Samples were incubated at 65°C for 1 hr in a Thermomixer with intermittent shaking (2000 RPM for 1 min every 15 min). 400 µL of ethanol was then added and the RNA was purified using the Direct-zol RNA Miniprep Plus kit (Zymo research) according to the manufacturer's protocol including the optional DNase digestion step. RNA integrity was confirmed using an Agilent Bioanalyzer.

## RNA-seq library preparation
RNA-seq libraries were prepared first by removing rRNA with the Ribo-Zero Gold rRNA Removal Kit (Yeast) (Illumina Cat #MRZY1324) and eluted in 30 µL. After rRNA removal, RNA was bound to Agencourt RNAClean XP beads (Beckman coulter cat#A66514) using a 2x bead volume (60 µL) followed by washing and elution as per vendor's protocol (except that RNA was eluted in Fragment,

Prime, Finish Mix from the TruSeq Stranded mRNA Library Prep [Illumina cat# 20020595]). Subsequent steps were carried out as per the protocol for the TruSeq Stranded mRNA Library Prep. Libraries were indexed using Illumina barcode kit (catalog # 20020492, 20020493 or 20019792). Libraries were sequenced on a hISEQ4000 (with $150 \times 150$ paired-end reads).

## RNA-seq data processing and analysis

RNA-seq data was quantified using Salmon-0.8.1 with three separate annotation indices:

1. open-reading frames (ORFs) as defined by SGD
2. 'complex transcriptome' index created by aggregating ORFs, their longest annotated UTRs + all non-canonical and non-coding transcripts pulled from SGD (references)
3. all features index composed of ORFs + non coding transcripts (SGD all freatures)+genomic intergenic bins

Salmon was run using the 'quant' command with the following settings:'libType': 'ISR', 'useVBOpt': [], 'numBootstraps': '30', and 'incompatPrior': '0'.

Salmon quantification was prepared and processed for differential expression analysis with the Wasabi package in R for input into Sleuth. Genes changing with respect to time (generation) as opposed to batch or strain were defined as significant using: model =~generation*strain+(batch)+(0)' and using a q < 0.05 for calculated beta values. For determining genes changing significantly across time (age) for each specific strain the model '~batch + strain +strain:generation' was specified. All test tables output from Sleuth included in *Source Data 1* and *Source Data 2*.

Batch correction (for heatmap visualization only) was applied to the Salmon quantification output matrix with a pseudocount of 2 TPM with the COMBAT package in R (*Johnson et al., 2007*). Hierarchical clustering (average linkage) was performed on the batch-corrected values using the uncentered Pearson correlation as the distance metric.

Gene Ontology (GO) term enrichment analysis was performed via uploading query lists to the DAVID Bioinformatics Resources 6.8 website and running enrichment with the *S. Cerevisiae* species background.

## Live/dead cell prep for ATAC-seq

WT cells (DBY12000) were grown in YPD to $OD_{600}$ 0.2–0.4, spun down (1200 rcf/4 min./room temperature), resuspended at $10^8$ cells/mL in YPD and split into two 1 mL aliquots for 'live' and 'dead' cells. The 'dead' cell aliquot was incubated at 50°C for 25 min in a thermomixer and 100 µL (10 million cells) was plated on YPD to test viability (only 2 CFUs grew). Live and dead cells were combined into 100 µL aliquots at the desired ratios. ATAC-seq was performed as described below.

## ATAC-seq sample preparation

5 million cells were spun down (1200 rcf/4 min./room temperature) and resuspended in 250 µL media. Propidium Monoazide (PMA, Qiagen 296015) was added to a concentration of 50 mM, the cells were incubated in the dark at room temperature for 10 min. and then illuminated in the BLU-V system (Qiagen 9002300) for 20 min at 4°C. Cells were spun down, washed twice with SB buffer (1M Sorbitol, 40 mM HEPES [pH7.5], 10 mM $MgCl_2$) and then resuspended in 190 µL SB Buffer. 10 µL of 10 mg/mL Zymolyase-100T was added and the cells were incubated at 30°C for 30 min. with light shaking (600 RPM) in a thermomixer (Eppendorf). Spheroplasts were spun down (1500 rcf/5 min./room temperature) and washed twice with SB Buffer, resuspended in 50 µL Tagmentation Mix (25 µL Nextera Tagment DNA Buffer, 22.5 µ L $H_2O$, 2.5 µL Nextera Tagment DNA Enzyme I) and incubated at 37°C for 30 min. DNA was purified over DNA Clean and Concentrator-5 kit (Zymo Research) following the manufacturer's protocol, eluted in 11 µL $H_2O$ and stored at $-20$°C until ready for PCR.

PCR reactions were set up using Nextera Index i5 and i7 series PCR primers (Illumina). Mixed 25 µL NEBNext Hi-Fidelity 2x PCR Master Mix, 7.5 µL $H_2O$, 6.25 µL i5 primer (10 mM), 6.25 µL i7 primer (10 mM) and 5 µL tagmented DNA from above such that each sample has a unique barcoded primer pair. Ran PCR amplification (one cycle: 72°C for 5 min.; one cycle: 98°C for 30 s.; eight cycles: 98°C for 10 s., 63°C for 30 s., 72°C for 1 min.; hold at 4°C). PCR reactions were cleaned up using the Agencourt AMPure XP system (Beckman Coulter) first by negatively selecting against large DNA fragments using 0.4x volume of beads and then by positively selecting for the desired fragments using

1.5x volume of beads. The final solution was re-purified over DNA Clean and Concentrator-5 columns to eliminate primers, eluted in 22 μL $H_2O$ and analyzed on the BioAnalyzer.

## ATAC-seq data analysis

### Processing of raw sequencing data

Raw ATAC-seq data was processed as described in *Hendrickson et al. (2018)*. Briefly, first, the raw paired end reads were aligned to the Saccharomyces cerevisiae genome (sacCer3, Release 64) using bwa mem version 0.7.12 with default parameters. Second, the alignments were filtered requiring that both reads in the pair are mapped, the mapping quality is greater than or equal to 30, the reads are mapping concordantly and the direction of the mapping is F1R2. Third, we converted the alignments into track of number of insertions at each position in the genome. For every aligned fragment, the insertion locations are calculated by shifting the fragment ends of reads that align to '+' strand by four nucleotides and shifting the fragment ends of reads that align to '-' strand by five nucleotides in the 3' direction to reflect the distance to the center of the transposase binding site (*Buenrostro et al., 2013*). To normalize for uneven sequencing depth and different amounts of rDNA in the samples, insertion counts were normalized such that the total number of insertions from unique non-rDNA, nonmitochondrial reads is constant in each sample.

### Annotations

Nucleosome positions were downloaded from *Brogaard et al. (2012)*, lifted over to sacCer3 assembly and only nucleosomes with mapping score greater than two were used in the metagene plots. For metagene plots around promoters we defined the TSSs for each yeast gene my merging the TSS data from *Pelechano et al. (2013)* (TIF-seq) and *Park et al. (2014)* (TSS data). If the TSS for a gene was not reported by these two papers, we used the beginning of ORF as the TSS. TF and chromatin modifier binding data were from (*MacIsaac et al., 2006*; *Venters et al., 2011*).

### NucleoATAC

Baseline estimation of the nucleosome occupancy from ATAC-seq data could be confounded by noise in estimation of nucleosome free and nucleosomal fraction of fragments between samples. Therefore we generated a fragment size distribution that was an average fragment size distribution of all samples and ran the NucleoATAC pipeline using a constant fragment size distribution using –sizes parameter. Since occupancy estimation was sensitive to the depth of the sample, we downsampled each sample to 25 million uniquely mapped non-ribosomal reads. Finally, to limit differences due to differences in nucleosomal calling between samples, we called occupancy only on the nucleosomes from Brogaard et al that are well positioned (score >2), located in the open chromatin part of the genome (as defined by Schep et al) and have a median occupancy of at least 0.5.

### Differential accessibility

Aging slope for accessibilities was calculated as follows. We first binned the genome into non-overlapping 100 bp bins and counted number of insertions in each bin. We then performed quantile normalization for each sample separately to calculate normalization factors and supplied gene specific normalization factors as described in http://bioconductor.org/packages/release/bioc/vignettes/DESeq2/inst/doc/DESeq2.html#sample-gene-dependent-normalization-factors to DESeq2 pipeline (*Love et al., 2014*). The aging slope was determined on all samples using the following model: '~replicate + strain+ ave_budscars' where ave_budscars were the average number of budscars for that time point and we looked at the bins with q-value of non-zero slope of less than $10^{-6}$. To identify bins that were specifically opening or closing in certain strains, we used the following model 'replicate +strain + ave_budscars + strain:ave_budscars'.

### Enrichments of regulators

We assigned each genomic bin to the gene whose gene body it intersects or to the promoter (defined as up to 400 bp upstream of TSS) of the gene that intersects. The lists of age-opening and age-closing genes were generated by looking at a genes that contained at least one opening and closing bin (with q-value $<10^{-6}$), respectively. We then intersected these lists with the list of genes

that bind to transcription factor or chromatin modifier and performed a hypergeometric test to evaluate enrichment.

## Strain construction

Strains containing single-gene deletions were constructed using the standard PCR-mediated gene disruption method (*Amberg et al., 2006*). NOP58-3xmNeonGreen fusion proteins were constructed in WT, *sir2Δ*, *ubr2Δ* and *fob1Δ* strains by PCR amplifying a 3xmNeonGreen/natMX cassette (from pKT127-3xNeonGreen-NatMX) using primers with homology to the NOP58 C-terminus and 3'UTR. All strains are S288c with a repaired *HAP1* allele (*Hickman and Winston, 2007*). A full list of strains used in this study can be found in *Figure 2—source data 2*.

## Nucleolar quantification

Strains were grown in media to mid-log phase ($OD_{600}$ of 0.2–0.4), labelled and beaded. Cells were harvested at log phase (pre-biotinylation), 'young' (purified after initial 20 hr of outgrowth, before loading onto MAD) and 'old' (after 23 hr of growth on the MAD, 43 hr of growth). Cells were stained with NeutrAvidin Protein, DyLight 405 (ThermoFisher cat# 22831) and Wheat Germ Agglutinin, Alexa Fluor 594 (ThermoFisher cat # W11262) to label the original biotinylated mothers and bud scars, respectively.

Approximately 50,000 cells/well were loaded onto a Matriplate 384-Well glass bottom microwell plate (Brooks Life Science Systems cat # MGB101-1-1-2-LG-L) which had been pre-treated for 10 min with 1 mg/mL concavalin A. Cells were imaged at 1000x magnification on a VT-iSIM microscope (BioVision Technologies).

Images were analyzed in ImageJ and nucleolar area was determined by first creating a mask outline around original (biotinylated, Neutravidin-405-stained) cells. This mask was overlaid onto the mNeonGreen image, the threshold of the green areas was set and the analyze particles function was applied to calculate the nucleolar areas within each original cell. If the nucleolus was fragmented, the area was recorded as the sum of all fragments within that cell. Nucleoli outside of cells or within non-neutravidin-405 stained cells were not included in the analyses.

## qPCR analysis of rDNA

PCR primers and probes were designed against each target sequence using the online PrimerQuest Tool (IDT-Integrated DNA Technologies) and ordered as PrimeTime QPCR Assays (primer and probe mix) with probes containing a 5' FAM reporter and ZEN/Iowa Black FQ double quencher. Each amplicon was also synthesized and cloned into a standard vector to generated copy # standard curves. Total DNA was isolated from frozen yeast pellets (5 million cells frozen in liquid nitrogen) via standard Zymolyase digestion/Buffered Phenol:Chlorofom:Isoamyl (25:24:1, pH 8.0) extraction/EtOH precipitation methods. Each amplicon plasmid was diluted to generate a standard curve ranging from 10 to 781,250 copies/well. qPCR was run on 250 pg of total DNA as well as the standard curve using IDT's PrimeTime Gene Expression 2x Master Mix in 384-well format on the Applied Biosystems ViiA seven system. Data was normalized to *ACT1* copy number to obtain relative copy number per genome. Primers can be found in *Figure 6—source data 1*.

## Accessing genomics data

RNA-seq and ATAC-seq data were deposited with the NCBI Gene Expression Omnibus with accession number GSE118581.

## Acknowledgements

We thank Jason Rogers, Anastasia Baryshnikova, Shelley Buffenstein, and David Botstein for their critical feedback on an earlier version of this manuscript. We'd also like to thank Nate Thayer, Manuel Hotz, Voytek Okreglak, and Dan Gottschling for many fruitful discussions.

## Additional information

### Competing interests

David G Hendrickson, Ilya Soifer, Bernd J Wranik, Griffin Kim, Michael Robles, Patrick A Gibney, R Scott McIsaac: Is affiliated with Calico Life Sciences. There are no other competing interests.

### Funding

| Funder | Author |
| --- | --- |
| Calico Life Sciences LLC | David G Hendrickson<br>Ilya Soifer<br>Bernd J Wranik<br>Griffin Kim<br>Michael Robles<br>Patrick A Gibney<br>R Scott McIsaac |

The funders had no role in study design, data collection and interpretation, or thedecision to submit the work for publication.

### Author contributions

David G Hendrickson, Conceptualization, Resources, Data curation, Software, Formal analysis, Investigation, Visualization, Methodology, Writing—original draft, Writing—review and editing, RNA-seq analysis; Ilya Soifer, Conceptualization, Resources, Data curation, Software, Formal analysis, Investigation, Visualization, Methodology, Writing—original draft, Writing—review and editing, ATAC-seq analysis; Bernd J Wranik, Data curation, Formal analysis, Investigation, Visualization, Methodology, Writing—original draft, Writing—review and editing, qPCR assay, Nucleolar imaging and quantification; Griffin Kim, Resources, Visualization, Methodology, Writing—original draft, Writing—review and editing, Wrote Supplementary File 1; Michael Robles, Methodology, 3D printing; Patrick A Gibney, Conceptualization, Methodology; R Scott McIsaac, Conceptualization, Supervision, Methodology, Writing—original draft, Project administration, Writing—review and editing

### Author ORCIDs

R Scott McIsaac (iD) http://orcid.org/0000-0002-5339-6032

### Decision letter and Author response

Decision letter https://doi.org/10.7554/eLife.39911.047
Author response https://doi.org/10.7554/eLife.39911.048

## Additional files

### Supplementary files

• Source data 1. RNA-seq data from dense aging time course.
DOI: https://doi.org/10.7554/eLife.39911.036

• Source data 2. RNA-seq data from aging times shown in *Figure 3*.
DOI: https://doi.org/10.7554/eLife.39911.037

• Source data 3. ATAC-seq data.
DOI: https://doi.org/10.7554/eLife.39911.038

• Source data 4. Transcription factors associated with genomic regions that change accessibility with age.
DOI: https://doi.org/10.7554/eLife.39911.039

• Supplementary file 1. MAD system setup manual
DOI: https://doi.org/10.7554/eLife.39911.035

• Transparent reporting form
DOI: https://doi.org/10.7554/eLife.39911.040

## Data availability

We've included all processed data in easily accessible tables. Sequencing data have been deposited in GEO under accession codes GSE118581

The following dataset was generated:

| Author(s) | Year | Dataset title | Dataset URL | Database and Identifier |
|---|---|---|---|---|
| Hendrickson DG, Soifer I | 2018 | Genomic analysis of aging yeast | https://www.ncbi.nlm.nih.gov/geo/query/acc.cgi?acc=GSE118581 | NCBI Gene Expression Omnibus, GSE118581 |

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

# Appendix 1

DOI: https://doi.org/10.7554/eLife.39911.041

We observed an acute response to the addition of the magnetic streptavidin coated beads that included an upregulation of the biotin biosynthesis pathway in *S. cerevisiae*, which includes enzymes and transporters involved in the synthesis and uptake of biotin and biotin precursors, respectively. We looked for and found this response in our data as well as in data from previous work reliant on the biotin/streptavidin interaction (*Appendix 1—figure 1*). We reasoned that the streptavidin beads were, in addition to binding the covalently linked biotin on the cell wall of labeled mothers, leaching biotin from the media and causing the upregulation of the biotin genes. Given that biotin is essential for yeast growth, we sought to test our aging protocol in a manner that would not perturb biotin levels by taking advantage of the fact that although *S. cerevisiae* cannot synthesize biotin de novo, it can make biotin from the precursor 7,8-diaminopelargonic acid (DAPA), which does not bind streptavidin (*Appendix 1—figure 1*). Furthermore, using DAPA would also enable beading in growth media.

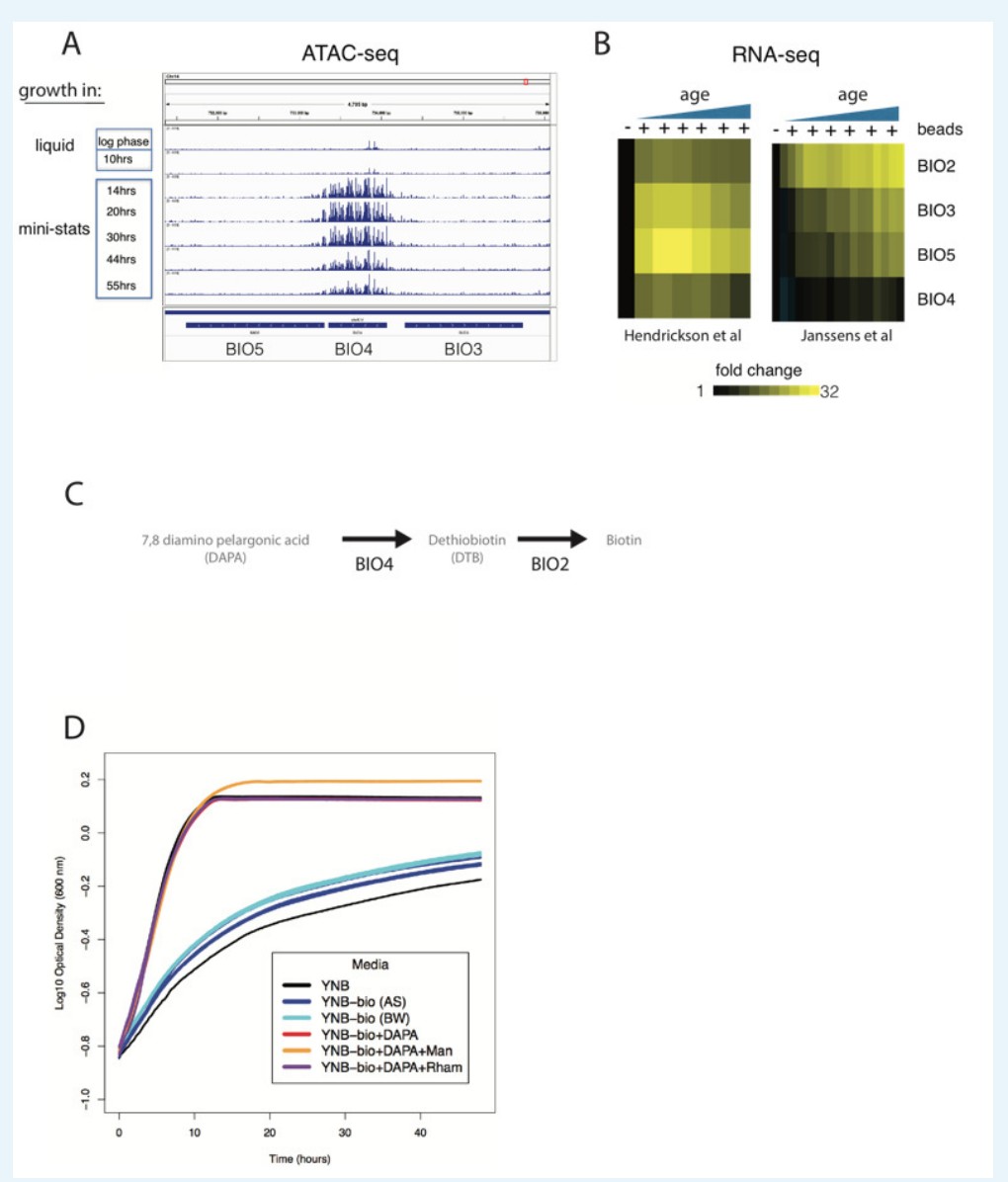

**Appendix 1–figure 1.** (**A**) ATAC-seq insertional density at the biotin gene locus of ministat-aged cultures. (**B**) RNA-seq analysis of biotin genes at increasing ages from this manuscript and Janssens *et al.* (**C**) Biotin synthesis from DAPA in yeast. (**D**) Growth curves of DBY12000 in different media. AS and BW are two separate media preparations lacking biotin.

DOI: https://doi.org/10.7554/eLife.39911.042

We first determined that *S. cerevisiae* could grow on DAPA media without any biotin at a rate comparable to wild type (*Appendix 1—figure 1*). We next aged yeast mothers in DAPA media in the ministats and compared their age (in bud scars) as well as age-dependent transcriptome to those mother cells grown in standard biotin-containing media (*Appendix 1—figure 2*). The strength of the correlation between wild type cells grown in biotin versus DAPA led us to conclude that the aging process in both conditions is extremely similar (*Appendix 1—figure 2*). Growth in DAPA media allowed for beading of cells in actual growth media. In initial MAD iterations, binding of magnetic beads was done in PBS (*Figures 1* and *2*).

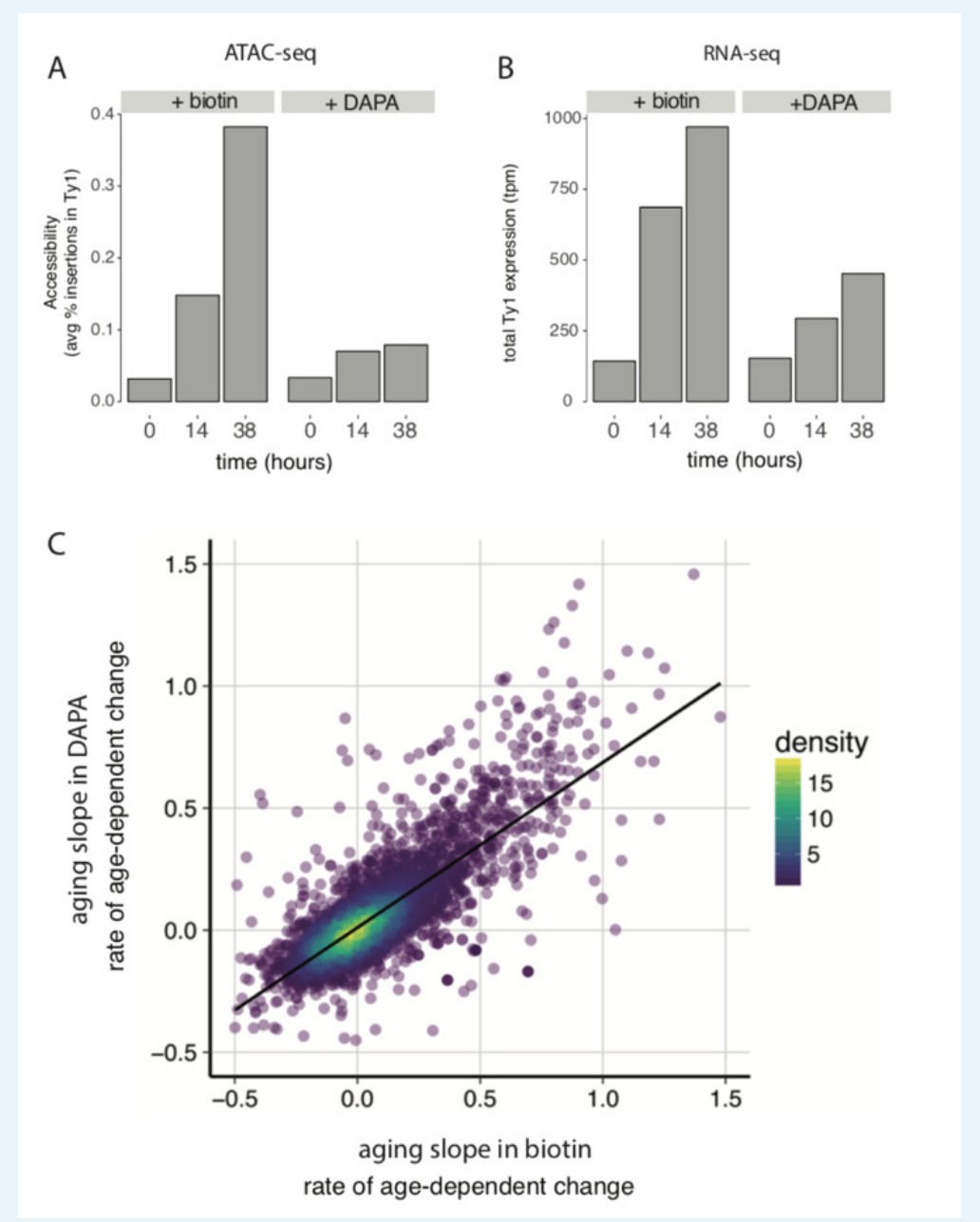

**Appendix 1–figure 2.** (**A**) ATAC-seq insertional density increases with age at Ty1 loci in both biotin and DAPA (**B**) Ty1 expression increases with old age in both biotin and DAPA. (**C**) Age-related gene expression changes are similar in media containing biotin or DAPA.
DOI: https://doi.org/10.7554/eLife.39911.043

