## [Decision Letter]

Thank you for submitting your article "A new experimental platform facilitates assessment of the transcriptional and chromatin landscapes of aging yeast" for consideration by *eLife*. Your article has been reviewed by two peer reviewers, including Matt Kaeberlein as the Reviewing Editor and Reviewer #1, and the evaluation has been overseen by Detlef Weigel as the Senior Editor.

The reviewers have discussed the reviews with one another and the Reviewing Editor has drafted this decision to help you prepare a revised submission.

Summary:

The reviewers appreciated the potential importance of this report both as a new method for studying yeast aging through enrichment for old mother cells and for the large genomics data sets that are generated. The MAD system provides an exciting, robust new tool that will be highly sought after by many labs across yeast aging field. The authors demonstrate that their new tool is highly capable, and facilitates robust acquisition of high quality data. For these reasons, the manuscript is well suited as a Tools and Resources article, and this will be an exciting advancement for the aging field. There were some significant concerns raised, however, about the lack of detail provided to allow replication by other groups and several places where the data appear to be over-interpreted. The authors are asked to carefully consider these comments and submit a revised version that (1) provides sufficient information on both the methodology used and the data generated and (2) addresses caveats and limitations to the data in the text. While some suggestions are made for additional lines of experimentation, these are not absolutely required, so long as the limitations to the current data set are appropriately noted. It is essential, however, that the procedures must be explained in sufficient detail that others with expertise would be able to build and operate the device, and a plan for sharing raw data must be explicitly included.

Essential revisions:

1) The method is not described in sufficient detail for other labs to replicate. The authors need to carefully go through their manuscript and ensure that a reasonably competent yeast lab would be able to construct and utilize one of their devices to reproduce the studies described here. They should include a new section with detailed instructions on how to build and run their system. This should include a list of all materials needed including vendors for non-ubiquitous pieces of lab equipment, technical details of operation (flow rates of air (not "minimum", but actual flow rates), etc.), relevant sizes of parts (size of magnets, diameter of tubing, etc.), as well as the optimization data that they must have collected to settle on their current operational methods (purity levels of mothers cells/doubling times, etc. for different sized magnets, different flow rates, different tube sizes, different starting cell counts, etc.) Videos would be optimal. Detailed information about the media used in each experiment are also important for others to be able to replicate these results or to understand potential sources of variation if some of these observations are not seen in other studies.

2) While the large data sets are impressive, the raw data appears to be lacking. Apparently, only spreadsheets where lists of significantly differently expressed genes or significantly opened/closed genes are presented. The authors should present expression levels or open/close for all transcripts/genes at all timepoints for all strains. There is no mention of uploading raw data to the GEO database. That should also be done to allow others to fully explore their data. Optimally the authors could put together a simple website where people can type in their gene of interest and then it outputs the results in a text table and a graph for all the different genotypes. At a minimum, all of the data needs to be uploaded somewhere, perhaps with a GitHub project to provide some basic scripts for processing.

3) There are several places where the authors make fairly strong claims that are not sufficiently supported by the data or by alternative methods of analysis. In these cases, the authors should either appropriately note the limited nature of the data or consider taking additional alternative approaches (e.g. single cell imaging by microfluidics, RT-qPCR, etc.). These include:

3A) Nucleosome occupancy with age ("Global nucleosomal occupancy does not decrease with age"). Although I agree that it's plausible that the nucleosome decrease with age is an artifact of dead cells like they discuss, I'm not so sure that they can make that claim so strongly. I really like their PMA method, but Figure 4C doesn't seem to quite so strongly support their claim. In particular, the 30h and 44h timepoints even with PMA look to have reduced nucleosome occupancy. This may or may not be significant, but they claim it isn't because the 55h timepoint looks like young cell occupancy again. It could be though, that subsets of cells age differently and that cells that lose nucleosome occupancy die younger, so the only old cells still alive by the last timepoint are those that don't experience nucleosome loss. Related to this, I would like to see them plot mean nucleosome occupancy in addition to median. On the single cell level the median cell at any timepoint looks "younger" than the mean because the old cells then die shortly thereafter and are removed from the population. I think that this could be happening with their nucleosome data.

3B) It was not clear if the authors were able to exclude the contaminating young cells reported in Figure 2—source data 1 from their RNA-seq and ATAC-seq analysis. If not, the high levels of contaminating young cells in the old cell samples is a bit concerning. If I am interpreting the results correctly, the very old cell populations analyzed by these methods contained 20-40% daughter cells. To confirm that their data isn't confounded by the newly developed method and/or daughter cell contamination, the authors could utilize parallel approaches to confirm hits from their RNA-seq and ATAC-seq datasets. For example, the authors could utilize a microfluidic device to analyze individual gene expression changes by monitoring fluorescence intensity of GFP being driven by a variety of promoters from genes identified as up or downregulated in the authors' datasets.

3C) Related to the previous point, the authors often draw conclusions from changes in RNA abundance, without confirming results with another approach. For example, the authors conclude based on RNA-seq data that the SRP is down in old cells. They then hypothesize, based on this observation and the increased expression of many secretory proteins and *SCR1*, that old cells likely have a problem with the SRP complex and ER protein translocation/translation efficiency. The authors could include additional assays to test this model. For example, are SRP subunit protein levels lower in aged cells by western blot or fluorescent microscopy? Can the authors detect differences in ER translocation ability in aged cells, or translation rates of individual proteins? Without confirming the SRP results with other techniques, it seems a bit premature to draw any conclusions about ER protein translocation in aging cells.

3D) The authors conclude that the large majority of gene expression changes in aged cells results from the ESR. While it is convincing that many genes known to be core ESR genes are altered in aged cells, the authors could test this hypothesis more directly by confirming that this transcriptional response is altered in mutant cells lacking transcriptional regulators of the ESR. Also, out of curiosity, are ESR mutants short lived?

3E) Sequencing the rDNA gives extremely variable results +/- 50% compared with CHEF gels. The authors make statements like "*sir2∆* cells had slightly more (~20%) relative rDNA copy number than WT at young ages (Figure 6A, B) but higher ATAC-seq coverage (2-fold), suggesting that during exponential growth, the difference between the strains is driven by bona fide differences in accessibility rather than by copy number." It would be more definitive to run a few CHEF gels to validate the sequencing results.

3F) The device or their technique seems to be unable to enrich cells above an average 19-20 divisions. This may contribute their inability to measure increased viability in aged long-lived strains and their inability to see more significantly bud scars in those long-lived strains. This seems troubling because one of their main conclusions is that the aging expression changes in long-lived strains is not much different that wild-type. Perhaps this is an artifact of their inability to see old cells.

---

## [Author Response]

Essential revisions:1) The method is not described in sufficient detail for other labs to replicate. The authors need to carefully go through their manuscript and ensure that a reasonably competent yeast lab would be able to construct and utilize one of their devices to reproduce the studies described here. They should include a new section with detailed instructions on how to build and run their system. This should include a list of all materials needed including vendors for non-ubiquitous pieces of lab equipment, technical details of operation (flow rates of air (not "minimum", but actual flow rates), etc.), relevant sizes of parts (size of magnets, diameter of tubing, etc.), as well as the optimization data that they must have collected to settle on their current operational methods (purity levels of mothers cells/doubling times, etc. for different sized magnets, different flow rates, different tube sizes, different starting cell counts, etc.) Videos would be optimal. Detailed information about the media used in each experiment are also important for others to be able to replicate these results or to understand potential sources of variation if some of these observations are not seen in other studies.

We thank the reviewer for the above comments, and we agree that more detail on how to build/run the system (including materials [with vendor information and specs]), as well as some data from optimization experiments are required. In the Materials and methods, we’ve added a new section called “Assembly of MADs” that includes much of the information listed above. We’ve added new supplemental figures (Figure 1—figure supplement 2 and Figure 1—figure supplement 3) that include renderings of 3D-printed posts and magnetic spacers complete with dimensional information, respectively. Figure 1—figure supplements 4 and 5 were added and include schematics of the bubble cage and custom caps, respectively. Most importantly, we have included the specific materials used for construction of each custom component of the system, as well as part numbers so that the custom components can be readily purchased by labs looking to implement this technology. An excerpt from the Materials and methods is shown below:

“A stainless steel (type SST316; materials from McMaster-Carr, 89495K425 and 9298K12) bubble cage (Zera Development, SBT001Rev01 and SBT002Rev01) is used to guide air bubbles to the center of the vessel so they don’t dislodge the bead-labeled mother cells from the glass wall (Figure 1—figure supplement 4). […] A Teflon cap (Zera Development, SBT003Rev01; materials from McMaster-Carr, 8546K17) with a silicone gasket (Zera Development, SBT004Rev01; materials from McMaster-Carr, 1460N25) is used for positioning of the needles and airtightness (Figure 1—figure supplement 5).”

As the reviewer suggested, flow rate information has been added to the Materials and methods (we typically run the devices at a dilution rate of ~25 mL/hr). We’ve also added the specific media formulations in the Materials and methods (including catalog numbers) and included additional optimization-related data in Figure 1—figure supplement 6 (which shows that keeping the bead-to-cell ratio under 20:1 is very important to not inhibit growth in the MAD). It is worth noting that the utilization of ring magnets, which allows the experimenter to distribute mother cells on the glass wall (so they don’t clump), is an important experimental detail that we have now included in the main text: “Ring magnets are an important component of this system because they enable an even distribution of magnetically-labeled mother cells along the glass walls of the ministat (i.e., they help minimize clumping).”

Finally, we’ve added an eleven page supplemental document (Supplementary file 1) that includes detailed instructions for building/operating the MAD system, as well as a Figure 1—source data 1, which includes ordering information for more than 45 components.

2) While the large data sets are impressive, the raw data appears to be lacking. Apparently, only spreadsheets where lists of significantly differently expressed genes or significantly opened/closed genes are presented. The authors should present expression levels or open/close for all transcripts/genes at all timepoints for all strains. There is no mention of uploading raw data to the GEO database. That should also be done to allow others to fully explore their data. Optimally the authors could put together a simple website where people can type in their gene of interest and then it outputs the results in a text table and a graph for all the different genotypes. At a minimum, all of the data needs to be uploaded somewhere, perhaps with a GitHub project to provide some basic scripts for processing.

We thank the reviewer for this comment, and we are in complete agreement: making both the raw and processed data useful/accessible to the community is important. To this end, we have uploaded all of the raw data to GEO (https://www.ncbi.nlm.nih.gov/geo/query/acc.cgi?acc=GSE118581). To the text, we’ve also added the statement “RNA-seq and ATAC-seq data were deposited with the NCBI Gene Expression Omnibus with accession GSE118581.” We have included the processed RNA-seq data for all genes and for all of the time courses shown in the paper (Dataset 1 and Dataset 2). For ATAC-seq, we decided to only include the bins with significant changes in Dataset 3 because there are literally millions of unchanging bins. We provide all of the ATAC-s data in bigWig format in GEO to make it easy for researchers to explore their favorite genomic regions in a genome browser such as IGV.

3) There are several places where the authors make fairly strong claims that are not sufficiently supported by the data or by alternative methods of analysis. In these cases, the authors should either appropriately note the limited nature of the data or consider taking additional alternative approaches (e.g. single cell imaging by microfluidics, RT-qPCR, etc.). These include:3A) Nucleosome occupancy with age ("Global nucleosomal occupancy does not decrease with age"). Although I agree that it's plausible that the nucleosome decrease with age is an artifact of dead cells like they discuss, I'm not so sure that they can make that claim so strongly. I really like their PMA method, but Figure 4C doesn't seem to quite so strongly support their claim. In particular, the 30h and 44h timepoints even with PMA look to have reduced nucleosome occupancy. This may or may not be significant, but they claim it isn't because the 55h timepoint looks like young cell occupancy again. It could be though, that subsets of cells age differently and that cells that lose nucleosome occupancy die younger, so the only old cells still alive by the last timepoint are those that don't experience nucleosome loss. Related to this, I would like to see them plot mean nucleosome occupancy in addition to median. On the single cell level the median cell at any timepoint looks "younger" than the mean because the old cells then die shortly thereafter and are removed from the population. I think that this could be happening with their nucleosome data.

We thank the reviewer for the above suggestion and agree that one of the caveats with bulk methods is our inability to disprove that particular cells that die at younger ages could do so through lost nucleosomes. In this paper we focused on examining a specific hypothesis: that the *average* nucleosomal occupancy of old cells is lower than in young cells. We provide evidence that at least a major part of this decrease is a result of increased concentration of the dead cells. We have modified our language accordingly, and the title of the section in the main text now reads “Evidence that global nucleosomal occupancy does not decrease with age.” As the reviewer suggested, we’ve included a new figure of mean occupancy versus time (Figure 4—figure supplement 3), which shows similar results to the median occupancy versus time.

3B) It was not clear if the authors were able to exclude the contaminating young cells reported in Figure 2—source data 1 from their RNA-seq and ATAC-seq analysis. If not, the high levels of contaminating young cells in the old cell samples is a bit concerning. If I am interpreting the results correctly, the very old cell populations analyzed by these methods contained 20-40% daughter cells. To confirm that their data isn't confounded by the newly developed method and/or daughter cell contamination, the authors could utilize parallel approaches to confirm hits from their RNA-seq and ATAC-seq datasets. For example, the authors could utilize a microfluidic device to analyze individual gene expression changes by monitoring fluorescence intensity of GFP being driven by a variety of promoters from genes identified as up or downregulated in the authors' datasets.

We thank the reviewer for this comment, and we agree – one of the issues at the very late time points is the increased presence of daughter cells in the samples. To address this, we collected pure daughter cell fractions from the effluent of one of the MAD runs and performed RNA-seq. As now shown in Figure 2—figure supplement 2A, the daughter cell gene expression signature cannot account for the majority of the aging signature we see in the MAD. Interestingly, there is a cluster of genes that gets repressed with age and then returns to baseline. While the repression is clearly age-associated, we can now attribute the nonlinearity in the response (i.e., the return to baseline) to contaminating daughter cells. We’ve added the underlined text to the Results section: “We found that a feature of the aging transcriptome is the activation of the ESR (i.e., almost all ESR-induced genes were upregulated with age and almost all ESR-repressed genes were down-regulated [Figure 2]). This signature, even at later time points (where the purity of mother cells decreases), cannot be explained by newly produced daughter cells (Figure 2—figure supplement 2A).”

3C) Related to the previous point, the authors often draw conclusions from changes in RNA abundance, without confirming results with another approach. For example, the authors conclude based on RNA-seq data that the SRP is down in old cells. They then hypothesize, based on this observation and the increased expression of many secretory proteins and SCR1, that old cells likely have a problem with the SRP complex and ER protein translocation/translation efficiency. The authors could include additional assays to test this model. For example, are SRP subunit protein levels lower in aged cells by western blot or fluorescent microscopy? Can the authors detect differences in ER translocation ability in aged cells, or translation rates of individual proteins? Without confirming the SRP results with other techniques, it seems a bit premature to draw any conclusions about ER protein translocation in aging cells.

We agree with the reviewers, and intended to point this out as an example of a hypothesis that could be generated from our data to showcase the utility of the MAD, rather than as a result. In the results, we now highlight the observations we made from analyzing our genomics datasets: that SRP clients tend to increase expression with age; that SRP subunits tend to decrease in expression in age; that we see increased *SCR1* RNA at older ages; and that ER stress components tend to increase in expression with age. We also performed an informatic analysis of the deleteome dataset from Holstege (Cell, 2014) and found that the negative correlation between expression of SRP components and SRP clients can be found there as well. We have rewritten the section to emphasize these points.

3D) The authors conclude that the large majority of gene expression changes in aged cells results from the ESR. While it is convincing that many genes known to be core ESR genes are altered in aged cells, the authors could test this hypothesis more directly by confirming that this transcriptional response is altered in mutant cells lacking transcriptional regulators of the ESR. Also, out of curiosity, are ESR mutants short lived?

We thank the reviewers for the above comment, which encouraged us to perform a literature search. We found one paper showing that *msn2∆msn4∆* strains are actually not short-lived on rich medium. However, an intact ESR is required for lifespan extension in conditions where glucose is lowered from 2% to 0.5%. Our current view is that the ESR is a consequence, rather than a cause, of aging. We’ve included these findings in the Discussion:

“A previous study provided evidence that deleting both of the primary ESR-related transcription factors, *MSN2* and *MSN4*, does not reduce RLS on rich medium (where all of the amino acids are provided); it was also found that these same transcription factors are required for lifespan extension under conditions of lowered glucose [Medvedik et al., 2007]. […] Currently, we favor a model in which the concordance we observe results from a cascade of cellular decline that results in age-associated slowing of growth, which is consistent with reports of age-dependent reductions in single-cell budding rates measured in microfluidics devices [Jo et al., 2015].”

3E) Sequencing the rDNA gives extremely variable results +/- 50% compared with CHEF gels,. The authors make statements like "sir2∆ cells had slightly more (~20%) relative rDNA copy number than WT at young ages (Figure 6A, B) but higher ATAC-seq coverage (2-fold), suggesting that during exponential growth, the difference between the strains is driven by bona fide differences in accessibility rather than by copy number." It would be more definitive to run a few CHEF gels to validate the sequencing results.

We thank the reviewer for this comment, and we have decided to remove the above statement from the text. We have re-written the above paragraph to read, “We looked at rDNA ATAC-seq coverage in *fob1∆* and *sir2∆* mutants, which repress and promote ERC formation, respectively. […] However, in looking at age-related increases at this locus in *sir2∆,* relative rDNA copy number increased by ~10-fold and ATAC-seq signal only increased by twofold at older ages; one interpretation of these data is that an initial open chromatin state may promote instability/higher copy number with age.”

Additionally, we have shortened the next section, and conclude with the following: “Compared to young cells, old cells from all genotypes showed nucleolar enlargement (Figure 7). However, the observed increases were clearly more pronounced in WT and *sir2∆* than in *ubr2∆* and *fob1∆*, similar to what was seen in the qPCR and ATAC-seq assays.”

One goal with these three experiments (qPCR, ATAC-seq, cell biology) is to show that we get similar results across very different data types and to demonstrate one of MAD’s most useful features: the ability to observe the segregation patterns (at the genomic level) between mother cells and their daughters (Figure 6).

3F) The device or their technique seems to be unable to enrich cells above an average 19-20 divisions. This may contribute their inability to measure increased viability in aged long-lived strains and their inability to see more significantly bud scars in those long-lived strains. This seems troubling because one of their main conclusions is that the aging expression changes in long-lived strains is not much different that wild-type. Perhaps this is an artifact of their inability to see old cells.

We thank the reviewer for this comment, and while the age-dependent changes are very similar, we do see some differences; for example, in *ubr2∆*, we see chronic activation in the gene expression of proteasome genes. However, the most dominant feature in both *fob1∆* and *ubr2∆* cells is increased rDNA stability, which we confirmed with three independent assays. It is worth pointing out that while the median ages at later time points were ~20 divisions, we were able to purify cells that were quite a bit older (25-30 divisions). That being said, we agree that the similar aging expression profiles might result, in part, from (1) the ESR dwarfing more subtle expression differences, (2) the large variance in ages at later time points and (3) an inability to get pure populations of the very oldest cells, which may reveal distinct expression signatures.

We’ve added the following sentence to the last paragraph of the Results section: “Additionally, technical improvements (such as being able to directly conjugate beads to yeast cell walls or reducing the variance in the ages of purified cells) could better facilitate detection of temporal differences between mutant strains, especially at older ages.”